# Reasoning Cache: Continual Improvement Over Long Horizons via Short-Horizon RL

**Ian Wu** [1]   **Yuxiao Qu** [1]   **Amrith Setlur** [1]   **Aviral Kumar** [1]

## Abstract

Large Language Models (LLMs) that continue improving at test-time budgets far beyond their training budgets can solve harder problems by leveraging additional inference compute: we refer to this property as *extrapolation*. Standard on-policy RL operates on fixed problem distributions and training budgets, giving rise to a distribution shift between train and test that limits the resulting model's extrapolation capabilities. To address this, we introduce `RC`, an iterative decoding algorithm replacing standard autoregressive decoding that enables models to extrapolate to lengths an order of magnitude longer than those seen during training. `RC` exploits the asymmetry between summarization and generation capabilities present in LLMs to construct a decoding process that improves consistently over iterations. Its effectiveness can be further increased through training, which amplifies the model's ability to perform summary-conditioned reasoning while avoiding the challenges of long-horizon RL. Empirically, training a 4B instruction-following model with `RC` using a 16k-token training budget improves performance on HMMT 2025 from 40% to 70% when evaluated with a 512k-token test budget, substantially surpassing comparably sized LLMs. Our code is publicly available at github.com/IanYHWu/rc.

## 1. Introduction

Large language models (LLMs) exhibit the ability to solve complex problems by generating long reasoning traces at test time. As LLMs become more capable, we naturally expect them to be able to solve harder tasks by reasoning for longer, even without external supervision. This expectation mirrors human cognition: humans improve their reason-

[1]Carnegie Mellon University, Pittsburgh, Pennsylvania, USA. Correspondence to: Ian Wu <ianwu@andrew.cmu.edu>.

*Proceedings of the 43rd International Conference on Machine Learning*, Seoul, South Korea. PMLR 306, 2026. Copyright 2026 by the author(s).

ing by revisiting earlier conclusions and reallocating effort to discover new information over the course of problem-solving. We would like our models to behave similarly, so that additional reasoning at test time translates to improved performance. This underlies the notion of *in-context exploration* (Setlur et al., 2025), which learns an implicit algorithm for allocating test-time compute such that spending more computation systematically improves outcomes. If learned robustly, in-context exploration should enable models to improve over long horizons at test time, perhaps over hours or even days, and across millions of tokens.

Current training paradigms limit the forms of in-context exploration that can be learned. Supervised fine-tuning (SFT) teaches models to imitate the content of reasoning traces, rather than the algorithm that generates them (Chu et al., 2025). This inhibits the learning of systematic reasoning "procedures" and thus in-context exploration. Reinforcement learning (RL) does better by incentivizing models to learn reasoning procedures rather than imitation (Sun et al., 2025; Zhang et al., 2025). However, RL still occurs only over fixed prompt distributions and within bounded training budgets (i.e., rollout length), resulting in models that are optimized only to utilize this budget rather than to *extrapolate* beyond it. When such models encounter complex problems that require more reasoning to solve, two failure modes emerge. First, they may prematurely terminate within their training budgets and fail to utilize available test-time compute even when more reasoning is beneficial. Second, when models continue beyond this budget, *distribution shift* may occur as generation proceeds from conditional distributions that differ substantially from those encountered during training; prior work finds that reasoning traces in this regime are often repetitive and verbose, leading to degraded performance (Luo et al., 2025; Setlur et al., 2025). This raises a natural question: **how can we *train* models to extrapolate their reasoning far beyond their training configurations?**

To address this, we introduce ***Reasoning Cache* (`RC`)**, an iterative decoding algorithm that replaces standard autoregressive decoding. In `RC`, the model generates a reasoning trace, summarizes it (the "cache") and discards the original trace over multiple turns, with subsequent reasoning conditioning only on the previous summary rather than the full history. Our approach is motivated by two ideas. First, itera-

*Figure 1.* **Left:** **Illustration of the RC algorithm.** RC decoding replaces standard autoregressive decoding at both train and test time. During RC decoding, the LLM generates a reasoning trace, summarizes it, discards the original trace, and conditions subsequent reasoning on this summary. This decouples the effective reasoning horizon from the length of any single reasoning trace, maintaining tractable rollout lengths for RL while also enabling extrapolation at test time. **Right:** **Performance on HMMT 2025 (November) vs. reasoning token budget.** Our RC-trained model RCT-4B (blue, trained from Qwen3-4B-Instruct-2507 at 16k train budget) extrapolates to outperform both the base model with RC decoding (green) and the specialized Qwen3-4B-Thinking-2507 reasoning model (evaluated at 256k test tokens).

tive decoding methods provide natural control over test-time compute by allowing us to scale the number of iterations while also maintaining bounded context lengths at each step. This keeps generation within the training distribution even as effective reasoning horizons grow, mitigating distribution shift. Second, making iterative decoding effective requires enabling progress across iterations, which RC achieves by exploiting *summarization-generation asymmetry*: that models are better at summarization and reasoning from summaries than generating correct solutions from scratch. In addition to being effective at test time, we also show that the structure of RC enables us to train models to extrapolate via RL. This training is amenable to off-policy learning via a *replay buffer* that enables reuse of cached summaries, allowing the model to train over long effective horizons without generating prohibitively long training rollouts.

Empirically, we find that models trained with RC exhibit consistent extrapolation behavior, and that RC is most effective when the base model can reliably follow instructions and build upon summaries. On the mathematical reasoning benchmarks HMMT 2025 and IMO-AnswerBench, our trained model substantially outperforms the base model through extrapolation. For example, on HMMT 2025, RC training improves a 4B instruction-following model from 40% accuracy at 16k tokens to 70% accuracy at 512k tokens (Figure 1) while on IMO-AnswerBench, performance improves from 41% accuracy to 58% at 256k tokens, surpassing larger models such as Nemotron-3-Nano-30B-A3B, despite training only on a 16k token budget. Moreover, our model, trained only on mathematical reasoning data, achieves far higher performance on the FrontierScience scientific reasoning benchmark than the base model, which suggests that RC transfers general algorithmic behavior rather than domain-specific knowledge alone.

## 2. Related Work

**Test-time extrapolation of reasoning.** Prior work attempt to enable extrapolation through two main approaches. The first modifies training via carefully designed datasets and

curricula to encourage in-context exploration (Setlur et al., 2025; An et al.; Luo et al., 2025). Although this enables extrapolation to 3–4× the training budget, performance typically saturates. The second approach instead modifies the RL reward structure to implicitly optimize performance beyond the training budget by introducing dense rewards that credit segments based on their contribution to overall progress (Qu et al., 2025b). In both cases, in-context exploration behaviors are implicitly learned through free-form autoregressive generation and remain coupled to the training setup. When test-time conditional distributions fall outside the training support, these approaches do not improve further and result in verbose and repetitive behavior (Setlur et al., 2025; Luo et al., 2025). These failures stem from the distribution shift between conditional distributions seen during train and test time, which our work aims to fix.

**Iterative decoding for scaling test-time compute.** Prior work explores prompting LLMs to iteratively transform their own outputs to scale test-time compute, ranging from simple self-correction (Huang et al., 2024b; Kim et al., 2023) and self-refinement (Madaan et al., 2023; Shinn et al., 2023) to more complex scaffolds that combine iterative and parallel compute (Shao et al., 2025). Others consider training models to apply transformations (e.g., aggregation (Venkatraman et al., 2025), self-correction (Qu et al., 2024; Kumar et al., 2024)) rather than relying on prompting alone. RC instead uses iterative decoding to enable extrapolation beyond training horizons rather than only improving performance at fixed budgets. As we show, RC is thus compatible with and can improve the performance of test-time scaffolds.

**Memory for multi-turn interaction.** RC summaries can be viewed as compressed memory states that are updated as the policy acts over iterations. Prior work primarily consider using similar memory states to store external context (e.g. retrieved web pages, user responses etc.) that is dynamically recalled in later steps, often as part of multi-turn question-answering or conversation systems (Li et al., 2023; Zhou et al., 2025). Our work instead uses the memory states to store self-generated reasoning traces for solving reasoning

problems, a setting also considered in Suzgun et al. (2025). Unlike their work, we focus on training the model to better utilize this memory, which we show yields significant improvements over prompting-only approaches.

## 3. Preliminaries and Notation

Consider a policy $\pi_\theta(\cdot|\mathbf{x})$ over token sequences, where generation occurs autoregressively conditioned on $\mathbf{x}$. At test time, the model is given a token budget and allocates this to reason. Our main interest is test-time performance as a function of the test-time token budget, particularly in regimes where test-time budgets exceed training budgets.

**Standard RL training for LLM reasoning.** Let $\mathcal{D}_{\text{train}}$ denote a training distribution of prompt-final answer pairs $(\mathbf{x}, \mathbf{y})$. On-policy reinforcement learning (RL) optimizes the expected reward of rollouts sampled from the model:

$$\max_{\pi_\theta} \ \mathbb{E}_{\mathbf{x},\mathbf{y}\sim\mathcal{D}_{\text{train}}} \left[ \mathbb{E}_{\mathbf{z}\sim\pi_\theta(\cdot|\mathbf{x})}[r(\mathbf{y},\mathbf{z})] \right],$$

$$\text{s.t. } |\mathbf{z}| \leq H_{\text{train}}. \quad \text{(Training objective)} \quad (1)$$

Here, $\mathbf{z}$ denotes an on-policy rollout autoregressively sampled from $\pi_\theta$. The rollout encodes a reasoning trace and is generated within a fixed training budget $H_{\text{train}}$. The reward function $r(\mathbf{y}, \mathbf{z})$ evaluates the correctness of the rollout, typically by extracting the final answer from $\mathbf{z}$ and comparing it against the ground-truth label $\mathbf{y}$. To solve this optimization problem, we can use outcome-reward policy-gradient methods: one common choice is GRPO (Shao et al., 2024) (see Appendix I), which we use throughout this work.

**Test-time extrapolation of LLM reasoning.** Equation 1 optimizes performance only over the empirical distribution of training prompts $\mathcal{D}_{\text{train}}$, and only under a fixed $H_{\text{train}}$. At test time, however, we may wish to maximize accuracy under a different distribution and under a larger budget:

$$\text{TestPerf}(\pi_\theta) \overset{\text{def}}{:=} \mathbb{E}_{\mathbf{x},\mathbf{y}\sim\mathcal{D}_{\text{test}}} \left[ \mathbb{E}_{\mathbf{z}\sim\pi_\theta(\cdot|\mathbf{x})}[r(\mathbf{y},\mathbf{z})] \right],$$

$$\text{s.t. } |\mathbf{z}| \leq H_{\text{test}} \quad \text{(Test-time objective)} \quad (2)$$

where $H_{\text{test}}$ is the test budget: in general, the training and test distributions differ (i.e. $p_{\text{train}}(\mathbf{x}) \neq p_{\text{test}}(\mathbf{x})$ and $H_{\text{test}} \gg H_{\text{train}}$). When a model trained to optimize Equation 1 can leverage a larger test budget to achieve $\text{TestPerf}(\pi_\theta)|_{H_{\text{test}}} > \text{TestPerf}(\pi_\theta)|_{H_{\text{train}}}$, we say that it *extrapolates*.

## 4. Problem Statement

Does optimizing train performance at $H_{\text{train}}$ (Equation 1) also optimize extrapolation at test budget $H_{\text{test}}$ (Equation 2)? Unfortunately, the answer is no. During training, the model receives positive reward only for rollouts that terminate within $H_{\text{train}}$ tokens. This implicitly penalizes longer trajectories and encourages *premature termination* near $H_{\text{train}}$ at test time. Moreover, when the model does continue beyond $H_{\text{train}}$, it operates in regimes for which it was never trained,

inducing a *distribution shift* between the conditional distributions seen during training and those encountered at test time. Taken together, both problems suggest that standard RL cannot effectively train models to extrapolate.

**Why do we need extrapolation?** Given that we have control over model training, can we simply increase $H_{\text{train}}$ to match $H_{\text{test}}$? There are three problems with this approach. First, any new test distribution we encounter may always contain harder problems requiring $H_{\text{test}} \gg H_{\text{train}}$ to solve, so we need to train models that can adapt on-the-fly. Second, RL memory and compute costs scale aggressively with sequence length, making long-length RL prohibitively expensive. Third, training at large $H_{\text{train}}$ is only effective if $\mathcal{D}_{\text{train}}$ contains problems that actually require extended reasoning to solve, but collecting such problems at scale may be infeasible. These challenges indicate that we cannot just scale $H_{\text{train}}$, and instead must train models to extrapolate.

## 5. Extrapolation with the Reasoning Cache

Our goal is to address the dual problems of premature termination and distribution shift so that policies optimized at a fixed train budget can still extrapolate. Our idea is to replace autoregressive decoding with an iterative decoding algorithm $\text{Alg}(\pi_\theta; \mathbf{x})$. This algorithm leverages the structure of reasoning along with asymmetries present in LLMs to support long-horizon reasoning at test time while remaining amenable to training under a much smaller $H_{\text{train}}$. We begin by formalizing the key desiderata that $\text{Alg}$ should satisfy.

**Choosing an effective decoding algorithm.** An effective choice of $\text{Alg}$ must satisfy two desiderata. First, it should define an iterative procedure in which the number of iterations monotonically controls test-time compute, with each iteration operating on conditional distributions that remain close to those encountered during training. An $\text{Alg}$ that satisfies this desideratum avoids the two key problems associated with autoregressive decoding and standard RL that we described earlier: **(1)** by allowing the test-time token budget and actual token usage to be increased at will (simply by increasing the iteration limit), $\text{Alg}$ avoids premature termination; and **(2)** by only ever autoregressively generating sequences of up to $H_{\text{train}}$ within each iteration, $\text{Alg}$ minimizes any shifts in conditional distributions between train and test even though the effective reasoning horizon is much larger. Second, the algorithm should retain expressivity comparable to autoregressive decoding, such that each iteration can refine or extend prior reasoning and explore new directions. An $\text{Alg}$ that satisfies this will produce improvements across many iterations, thereby enabling extrapolation.

### 5.1. RC: A Multi-Turn Decoding Algorithm

We now introduce a decoding algorithm that satisfies these desiderata. Our algorithm, which we call **Reasoning Cache** (RC), is an iterative decoding approach that alternates be-

tween response generation and summarization. Being an iterative decoding algorithm, **RC** naturally fulfills our first desideratum: we can increase test-time compute by increasing the number of summarization-generation turns, while also avoiding significant shifts in the conditional distributions encountered at each turn by only ever autoregressively generating at bounded lengths $H_{\text{train}} \ll H_{\text{test}}$. To satisfy our second desideratum, **RC** relies on two properties of LLMs. First, reasoning traces are highly *redundant*: many tokens encode steps that are useful for local progress but need not be retained verbatim to guide future reasoning. This allows us to discard a significant portion of tokens (e.g. through summarization) so long as key information is retained. Second, many LLMs exhibit *summarization-generation asymmetry*, in that producing a correct response conditioned on a summary of a previous attempt is easier than generating a correct solution from scratch; this asymmetry arises from the instruction-following abilities of LLMs, which allows them to use summaries of prior generations to guide further reasoning. **RC** exploits this by periodically compressing reasoning into a cache and conditioning subsequent generation on it, allowing the model to refine, extend, or restart reasoning across iterations as needed. See Figure 14 for an example of **RC**'s outputs.

Let $\mathbf{x}$ denote the prompt and let $t \in \mathbb{N}$ index the decoding turn. **RC** maintains: **(1)** a reasoning trace $\mathbf{z}_R^{(t)}$ and **(2)** a summary $\mathbf{z}_S^{(t)}$, with $\mathbf{z}_S^{(0)}$ initialized to the empty string. At each turn, $\mathbf{z}_R^{(t)}$ is generated under a fixed token budget $H_R$, while $\mathbf{z}_S^{(t)}$ is generated under $H_S \ll H_R$. Decoding proceeds by alternately prompting the base model with two distinct system instructions $\mathcal{I}_R$ and $\mathcal{I}_S$ (see Appendix Q):

$$\mathbf{z}_R^{(t)} \sim \pi_\theta\left(\cdot \mid \mathcal{I}_R, \mathbf{x}, \mathbf{z}_S^{(t-1)}\right), \tag{3}$$

$$\mathbf{z}_S^{(t)} \sim \pi_\theta\left(\cdot \mid \mathcal{I}_S, \mathbf{x}, \mathbf{z}_R^{(t)}, \mathbf{z}_S^{(t-1)}\right). \tag{4}$$

$\mathcal{I}_R$ instructs the model to generate reasoning conditioned on the current cache, while $\mathcal{I}_S$ instructs the model to compress the current reasoning trace and previous summary into an updated summary that encodes high-level information about the strategies employed and conclusions reached in previous turns. After T turns, the final output is given by $\mathbf{z} := \mathbf{z}_R^{(T)}$. We denote this iterative process (see Figure 1) as:

$$\left(\mathbf{z}_R^{(1)}, \mathbf{z}_S^{(1)}, \ldots, \mathbf{z}_S^{(T-1)}, \mathbf{z}_R^{(T)}\right) \sim \text{Alg}(\pi_\theta; \mathbf{x}). \tag{5}$$

**Extrapolation with RC.** Because each step is allocated a fixed budget $H_R + H_S$, the total effective budget under **RC** is $T \times (H_R \times H_S)$. Since $H_R \gg H_S$, we approximate the budget as $T \times H_R \overset{\text{def}}{:=} H_{\text{train}}$. If performance improves when $T' \times H_R \gg H_{\text{train}}$, we say that **RC** enables extrapolation.

## 5.2. Experimental Evaluation

**Experimental setup.** We now validate whether LLMs possess the ability to to utilize **RC** without additional training. We evaluate **RC** decoding with Qwen3-4B-Instruct-2507 and Qwen3-30B-A3B-Instruct-2507 (Qwen Team, 2025), two LLMs capable of complex reasoning and instruction-following (see Appendix E for results from another model family). Using these LLMs, we run **RC** decoding for $T = 12$ turns with $H_S = 2048$ and $H_R = 16\text{k}$, yielding a total budget of $H_{\text{test}} = 192\text{k}$. This is far larger than the model's $H_{\text{train}}$, which we estimate to be about 16k (Appendix C). We use the November version of HMMT 2025 as our dataset, and generate 16 **RC** outputs per problem.

**Finding 1: RC enables extrapolation.** We plot how accuracy evolves with the token budget $H_{\text{test}}$ in Figure 2(a). We find that **RC** extrapolates reasoning far beyond $H_{\text{train}} = 16\text{k}$: 4B accuracy increases by 17% as the token budget is scaled from 16k to 192k, while 30B accuracy increases by 12%. We also plot actual token usage in Figure 2(b). We find that the cumulative tokens used scales linearly with the provided budget, which indicates that the model utilizes additional test time compute as provided. Overall, our finding demonstrates that **RC** enables effective extrapolation.

**Finding 2: Summary-based abstractions are key to effective extrapolation.** We study the role of summarization–generation asymmetry by removing the summary step and conditioning each iteration directly on the full reasoning trace $\mathbf{z}_R^{(t)}$, and prompt Qwen3-4B-Instruct-2507 ($H_R = 16\text{k}$) to either verify-then-correct (self-verify) or self-refine its solution (see Appendix N). Figure 2(c) shows that **RC** consistently outperforms these baselines across all $H_{\text{test}}$ values, demonstrating that summary-conditioned generation provides benefits beyond iterative reasoning alone. We attribute this to two factors. First, conditioning on summaries keeps the context length at each iteration bounded and aligned with the training distribution, whereas iterating on raw traces causes reasoning length to grow beyond $H_{\text{train}}$, inducing distribution shift. Second, raw reasoning traces contain redundant tokens that degrade performance as they accumulate by acting as "distractors" (Hong et al., 2025), while summaries distill the most important information and thus provide clearer guidance for subsequent reasoning.

**Finding 3: Summary detail level matters.** Next, we study how much information summaries should retain. We vary the prompt $\mathcal{I}_S$ to produce summaries of varying detail levels, ranging from answer-only to multiple paragraphs (Figure 20), and compare them to our default moderate summaries and full-trace conditioning. Figure 3 (left) illustrates the accuracy of Qwen3-4B-Instruct-2507 at $H_{\text{test}} = 192\text{k}$ for different $\mathcal{I}_S$. Accuracy degrades with very short summaries, improves with added detail, and peaks with two-paragraph summaries. Going beyond this degrades performance.

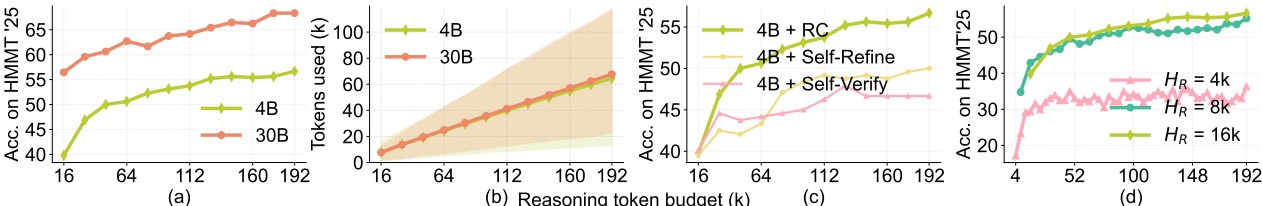

*Figure 2.* **(a) Accuracy vs. test-time token budget.** RC decoding improves performance as token budget $H_{\text{test}}$ is increased far beyond $H_{\text{train}} = 16$k. **(b) Total tokens used vs. test-time token budget.** Total reasoning tokens used by RC increases linearly as we increase the reasoning token budget. Shaded regions indicate the 5th–95th percentile; line indicates the mean. **(c) Accuracy vs. token budget for iterative decoding methods.** RC is a more effective method for enabling extrapolation than self-verification and self-refinement. **(d) Accuracy vs. token budget at various $H_R$.** Reducing $H_R$ beyond 8k negatively impacts RC performance.

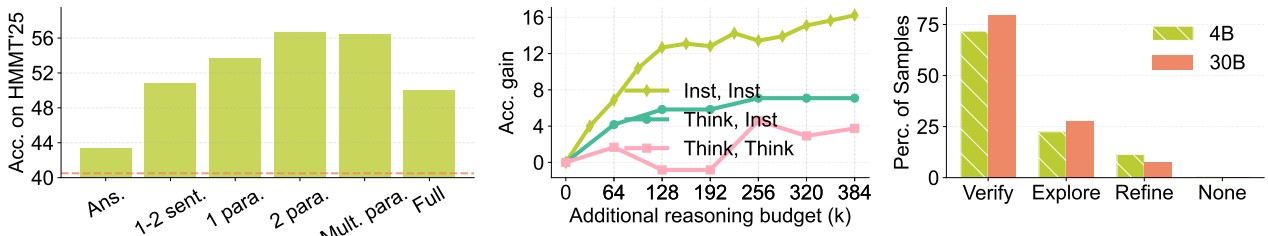

*Figure 3.* ***Left:*** **Accuracy at different levels of summary detail.** All accuracies measured at $H_{\text{test}} = 192k$, red dotted line indicates performance without RC. ***Middle:*** **Relative accuracy improvement (over standard autoregressive decoding).** Replacing instruct model with the thinking model for summarization (Think, Inst) reduces gains. Using the thinking model for both (Think, Think) further reduces gains. ***Right:*** **Percentage of RC reasoning traces employing various reasoning strategies.** *Verification* is the most common strategy.

**Finding 4: Base models must be good instruction-followers for RC to be effective.** We replace Qwen3-4B-Instruct-2507 with the specialist reasoning model Qwen3-4B-Thinking-2507, which excels at reasoning but possesses weaker instruction-following abilities and thus less summarization-generation asymmetry. We evaluate using the reasoning model only for summary-conditioned generation ("Think, Inst") and for both summary generation and summary-conditioned generation ("Think, Think") ($H_R = 64$k). Figure 3 (middle) shows that "Think, Inst" only achieves half the accuracy gains of "Inst, Inst", while "Think, Think" performs even worse; qualitative inspection reveals that the reasoning model often ignores summaries during generation and omits important details during summarization. This indicates that strong summarization-generation asymmetry is critical, motivating our focus on instruction-following models when training with RC.

**Finding 5: Reducing $H_R$ by too much degrades performance.** By default, we set $H_R = H_{\text{train}} = 16$k, and since LLMs rarely generate traces longer than $H_{\text{train}}$, we only consider decreasing it. Figure 2(d) shows that reducing $H_R$ to 8k has minimal impact despite the fraction of incomplete traces increasing from 0% to 20% (see Figure 9). Reducing $H_R$ to 4k causes nearly 50% of traces to terminate early, which substantially degrades performance. This implies that while $H_R$ can be decreased, it must still be large enough for redundancy to emerge. When $H_R$ is set too small, the resulting summaries capture only shallow progress that provides insufficient signal for continuation, which encourages

the model to restart reasoning from scratch instead.

### 5.3. Analysis of Summary-Conditioned Generations

We analyze the content of summary-conditioned generations produced by RC, and find that they commonly exhibit three high-level strategies: **(1)** *verification*, where the model generates reasoning to explicitly verify results in the input summary; **(2)** *exploration*, where the model explores a different strategy from that which was utilized in the summary, and; **(3)** *refinement*, where the model acknowledges the summary but repeats the same strategy without attempts to verify past logic or explore new strategies. To quantify which strategies dominate, we extract summaries and their subsequent reasoning traces and pass them to an LLM-based annotator (Figure 21), which assigns each sample to one of the three categories above (as well as a fourth, *none* category). From Figure 3 (right), we see that the LLM relies heavily on summaries to guide subsequent generations, with very few samples classified as *none*. The most common strategy is *verification*, although a substantial minority of samples also utilize *exploration* and *refinement*.

## 6. Training Models to Extrapolate with RC

Having established the design of Alg, we now describe our method for training models to use it. Our analysis in Section 5.2 shows that RC extrapolation depends on the model's ability to iteratively reason from and improve upon summaries of past reasoning. Accordingly, our training objective is to strengthen *summary-conditioned generation*:

given a problem and a summary, we should train our model to generate improved reasoning that is more likely to yield a correct answer. The iterative structure of RC makes this objective amenable to standard outcome-reward RL: we can run RC for multiple iterations and apply gradient updates independently at each turn, and by setting $H_R$ large enough that most iterations produce complete answers, we can assign outcome-based rewards at every step and thereby avoid the need to perform credit assignment across iterations. Formally, we run RC for $T_{\text{train}}$ turns for each problem $\mathbf{x}$ in a training batch. We collect the summaries generated from this $\mathbf{z}_S := (\mathbf{z}_S^{(1)}, \ldots, \mathbf{z}_S^{(T_{\text{train}})})$ and uniformly sample $N_{\text{summ}} \leq T_{\text{train}}$ unique summaries per problem. We then generate $K$ reasoning traces conditioned on each sampled summary, assign rewards based on correctness and compute advantages over these $K$ samples:

$$\max_{\pi_\theta} \quad \mathbb{E}_{\mathbf{x},\mathbf{y} \sim \mathcal{D}_{\text{train}}, t \sim U[1, T_{\text{train}}]} \left[ \mathbb{E}_{\mathbf{z}' \sim \pi_\theta(\cdot|\mathbf{x}, \mathbf{z}_S^{(t)})} \left[ r(\mathbf{y}, \mathbf{z}') \right] \right],$$
$$\text{where} \quad \mathbf{z}_S^{(t)} \sim \text{Alg}(\pi_\theta; \mathbf{x}), \quad \text{s.t.} \ |\mathbf{z}'| \leq H_R. \quad (6)$$

This design is effective because optimizing each turn locally generally also optimizes the full reasoning trajectory. The idea here is that by training every step $t$ to generate the correct answer, we also *implicitly* train the model to produce reasoning traces containing information (intermediate results, analysis etc.) that is more useful for subsequent steps. Furthermore, by conditioning training rollouts on these summaries, we train the model to better utilize this information, increasing the likelihood that subsequent steps produce correct answers. We are thus able to optimize for correctness over the entire trace despite only explicitly optimizing the correctness of steps individually. See Appendix F for a discussion on the limitations of this argument and for experiments considering alternative approaches.

**Training with a summary replay buffer.** The iterative decoding structure of RC naturally enables learning from off-policy summaries because summaries serve as conditioning inputs rather than optimization targets. Doing so provides two benefits. First, it enables training on later reasoning turns without the need for us to generate long on-policy trajectories, which is useful because summaries from later turns may qualitatively differ from earlier ones. Second, it increases the coverage of summaries the model encounters during training, which in turn increases the robustness of the policy to test time shifts in the summary distribution.

Motivated by this, we incorporate a *summary replay buffer* into our training algorithm; see Figure 10. During the first epoch of training, we follow the same procedure as before to optimize Equation 6, but also store all generated summaries, and their corresponding problems, in the replay buffer $\mathcal{B}$. From the second epoch onward, we sample problems and summaries from $\mathcal{B}$ and condition RC rollouts on them instead of generating fresh summaries, thereby extending the

maximum effective training horizon by $T_{\text{train}}$ per epoch.

# 7. Experimental Evaluation

**Training details.** We post-train a Qwen3-4B-Instruct-2507 model to utilize RC and refer to the resulting checkpoint as RCT-4B. We set $K = 8$, $N_{\text{summ}} = 2$ and $T_{\text{train}} = 3$ for training. We conduct training in two stages: in Stage I, we train without the summary replay buffer, focusing on optimizing early turns, including the initial turn (without any summary context). Training problems for Stage I are subsampled from the AceReason-Math dataset (Chen et al., 2025), resulting in a dataset of about 5.7k problems. For Stage II, we enable the summary replay buffer, and construct a new training set by injecting a small number of difficult problems from DAPO (Yu et al., 2025) into our Stage I dataset as part of our training curriculum. See Appendix J for details of our dataset construction procedure.

**Benchmarks and evaluation protocols.** We evaluate RCT-4B on three math reasoning benchmarks: **AIME 2025**, **HMMT 2025** (November version), and **IMO-AnswerBench** (IMO) (Luong et al., 2025), as well as one scientific reasoning benchmark, **FrontierScience** (FS) (OpenAI, 2025). Since our training data consists of math problems, FS serves to assess whether extrapolation behavior generalizes to an unseen domain. We evaluate the math benchmarks by verifying final answers and follow the official evaluation protocols for IMO (Luong et al., 2025) and FS (OpenAI, 2025). Note that HMMT 2025 (Nov), IMO, and FS were all released after our training data cutoff and after the release of our base model, significantly reducing the risk of contamination. See Appendix M for additional experiments extending RC to proof-based problems.

**Baselines.** We compare our trained model against three categories of baselines. The first consists of **autoregressive decoding** methods using open-source 4B models. These include Qwen3-4B-Instruct-2507 (our base model), Qwen3-4B-Thinking-2507, Polaris-4B (An et al.), a Qwen3-4B-based model trained for extrapolation by expanding the output context using YaRN (Peng et al., 2023), and a version of Qwen3-4B-Instruct-2507 trained with standard GRPO at $H_{\text{train}} = 32\text{k}$, the maximum output length we could reliably use for RL due to practical constraints. The second category of baselines consists of other **iterative decoding** approaches that condition directly on raw past reasoning rather than on summaries. We evaluate base and trained (see Appendix N) versions of two such methods: self-refinement and self-verification. We select $H_R$ and $H_{\text{test}}$ to be the same as for our RC experiments. The third category of baselines consists of approaches that do not train with RC but still use RC at inference time. We compare RCT-4B against inference-only use of RC by the base model (as in Section 5.2) and by the base model post-trained with standard RL. This comparison isolates the contribution of training with RC, rather

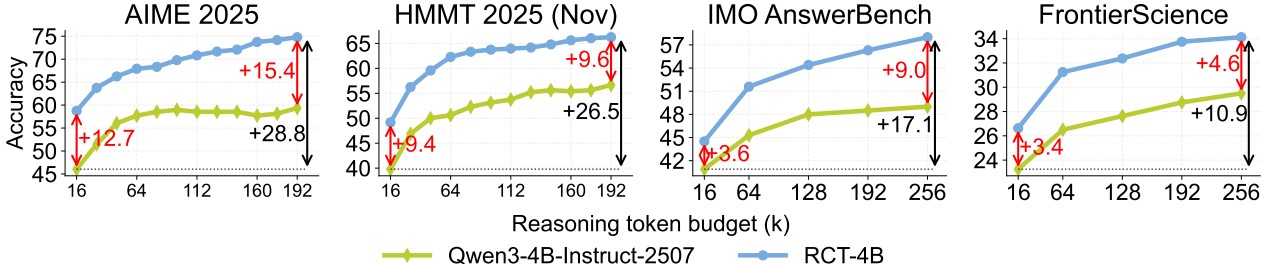

*Figure 4.* **Accuracy on various reasoning benchmarks as a function of token budget.** `RCT-4B` improves performance across all four benchmarks as token budget $H_{\text{test}}$ is increased beyond $H_{\text{train}}$. This improvement is larger than that attained by using the base model.

than applying it solely at test time. For discussion on the computational efficiency of **RC**, see Appendix G.

### 7.1. Benchmark Results

*Table 1.* **Evaluation results for baseline methods.** *Top:* standard autoregressive decoding with $H_{\text{test}}$ (in brackets). *Middle:* iterative decoding at 192k (AIME, HMMT) or 256k (IMO-AnswerBench, FrontierScience). *Bottom:* **RC** results using the same budgets. `RCT-4B` outperforms baselines on 3 out of 4 benchmarks.

|  | AIME | HMMT | IMO | FS |
|---|---|---|---|---|
| Base [16k] | 46.0 | 39.8 | 40.9 | 23.3 |
| Base (RL, 32k) [32k] | 54.8 | 48.3 | 42.1 | 21.5 |
| Polaris-4B [90k] | 79.4 | 60.2 | 47.7 | 23.6 |
| Thinking [81k] | **81.3** | 62.5 | 53.8 | 25.7 |
| Self-Refine (base) | 53.8 | 50.0 | 45.8 | 27.8 |
| Self-Verify (base) | 48.9 | 46.7 | 43.9 | 29.7 |
| Self-Refine (trained) | 60.4 | 61.3 | 52.2 | 33.5 |
| Self-Verify (trained) | 61.2 | 62.1 | 52.3 | 31.9 |
| Base + **RC** | 59.4 | 56.7 | 46.3 | 29.5 |
| Base (RL, 32k) + **RC** | 66.0 | 60.2 | 53.4 | 33.5 |
| **RCT-4B + RC** | 74.9 | **66.3** | **58.0** | **34.1** |

Our benchmark results are shown in Figure 4 and Table 1. Across all benchmarks and token budgets, `RCT-4B` outperforms the base model using **RC**, with the performance gap widening as the token budget increases. This indicates that training enables more effective extrapolation rather than merely improving short-horizon performance. Notably, the model also improves on FrontierScience despite being trained exclusively on math reasoning, suggesting that **RC** training develops domain-general extrapolation capabilities.

**RCT-4B is a highly competitive 4B model.** We compare `RCT-4B` against strong 4B reasoning-specialized models utilizing autoregressive decoding. While these models are explicitly trained to exploit large token budgets, `RCT-4B` outperforms all autoregressive baselines on the three benchmarks with lowest contamination risk: HMMT 2025 (Nov), IMO, and FS. In fact, we find that `RCT-4B` even achieves competitive results against much larger reasoning models: see Figure 7. Notably, while standard RL training improves upon the base model on mathematical reasoning benchmarks, our standard RL-trained model **(1)** remains substan-

tially weaker than specialized reasoning models, and **(2)** achieves no gains on the out-of-domain FrontierScience benchmark despite training on the same data as `RCT-4B`. This demonstrates that **RC** training develops generalizable problem-solving strategies that standard RL does not.

**RC training yields better iterative reasoners than other forms of iterative training.** `RCT-4B` substantially outperforms all base iterative decoding methods, including self-verification and self-refinement. Training the model to perform self-verification and self-refinement using a similar approach to **RC**-training (see Appendix N) yields models that substantially improve over their base models. However, these trained models are still considerably worse on mathematical reasoning tasks than `RCT-4B`, further demonstrating the benefits of exploiting the summarization-generation asymmetry through training. Interestingly, almost all of our iterative decoding baselines, including the untrained ones, outperform our autoregressive generation baselines on FrontierScience. This suggests that iterative decoding methods may generalize to out-of-domain problems better than standard long-horizon autoregressive decoding.

### 7.2. Evaluating **RC** on Hard Problems

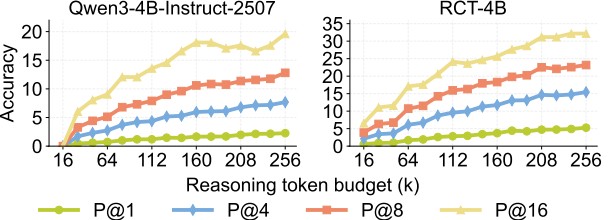

*Figure 5.* **Pass@$k$ accuracy vs. token budget on a hard subset of problems sampled from Qu et al. (2025a).** The *left* panel shows base model performance while the *right* panel shows `RCT-4B` performance. RCT-4B achieves higher pass@$k$, with the performance gap increasing as reasoning token budget grows (Figure 15).

Our results thus far show that extrapolation with `RCT-4B` yields consistent improvements on standard benchmarks. However, these gains could have arisen either from solving harder problems with test-time compute or from sharpening (Huang et al., 2024a) on problems that are already solvable within $H_R$. To distinguish between these effects, we

evaluate models on a set of adversarially curated problems from Omni-MATH (Gao et al., 2024): following Qu et al. (2025a), we select problems that Qwen3-4B-Instruct-2507 model fails to solve across $N = 256$ attempts.

We evaluate both the base model and RCT-4B on this dataset using **RC** decoding and report pass@$k$ in Figure 5. While both improve with increased token budgets, the gains for RCT-4B are substantially larger. At a budget of 256k tokens, the base model achieves a pass@16 of 20%, whereas RCT-4B reaches nearly 35%. Moreover, the performance gap between the models widens with increasing budget (Figure 15, right), indicating that training improves use of **RC**. Overall, our finding indicates that **RC** enables models to solve difficult problems that parallel compute alone cannot.

### 7.3. Ablation Studies

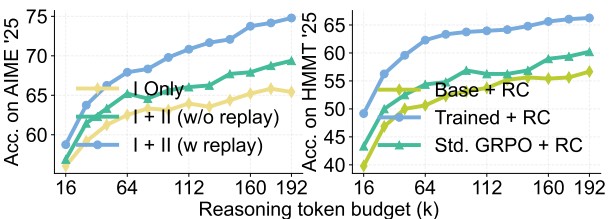

*Figure 6. Left:* **Ablation study on stagewise training configurations.** Performance improves with Stage II training, with replay buffers providing increasing gains at larger budgets. ***Right:* Comparison of training methods.** **RC**-specific training outperforms standard RL training, demonstrating that extrapolation requires explicit training beyond simply improving reasoning capabilities.

**Effect of the summary replay buffer.** We ablate our training procedure by comparing performance on AIME 2025 after **(i)** Stage I training, **(ii)** Stage II training without summary replay buffer, and **(iii)** Stage II training when the replay buffer is used; see Figure 6 (left). Stage I training alone yields gains over the base model, while Stage II training provides additional improvements. These gains are significantly larger with the replay buffer than without, particularly at higher budgets, indicating that using the replay buffer is critical for training models to maximally exploit **RC**.

**Importance of training with RC.** We isolate the effect of **RC** training by comparing RCT-4B against a model trained with standard RL. We train Qwen3-4B-Instruct-2507 with RL for the same number of steps over two stages, and evaluate it using **RC** decoding; see Figure 6 (right) and Table 1 (bottom). While RL training yields modest improvements, it falls far short of training with **RC**. This further demonstrates that extrapolation does not emerge from standard RL.

**Effects of varying $T_{train}$.** We vary $T_{train}$ during Stage I, evaluating $T_{train} \in \{2, 3, 4\}$; see Figure 15 (right). All settings improve extrapolation, but $T_{train} = 3$ performs best, followed by $T_{train} = 2$. When $T_{train}$ is too large, the model receives insufficient optimization on early turns, including

the initial unconditioned turn, which we find degrades performance. Conversely, using too small a value of $T_{train}$ limits exposure to summary-conditioned reasoning.

### 7.4. Incorporating **RC** into test-time scaffolds

*Table 2.* **RCT-4B gains from test-time scaffolds on HMMT 2025.** Using **RC** decoding yields further improvements.

|  | RSA | DSM Agent |
| --- | --- | --- |
| Base | 66.3 | 57.5 |
| RL | 64.8 | 61.3 |
| RCT-4B (no **RC**) | 70.2 | 65.8 |
| RCT-4B + **RC** | **75.4** | **74.6** |

We experiment with incorporating RCT-4B and **RC** decoding into two test-time scaffolds: RSA (Venkatraman et al., 2025) and DeepseekMath (DSM) Agent (Shao et al., 2025). The former iteratively generates and aggregates parallel reasoning traces, while the latter iteratively performs self-verification and self-refinement over an initial pool of solutions: see Appendix H for details. Our main goal is to understand whether RCT-4B and **RC** decoding can benefit from test-time scaling via agentic scaffolds. From Table 2, we see that RCT-4B is able to leverage both RSA and DSM Agent far more effectively than the base Qwen3-4B-Instruct-2507 model and its standard RL-trained version, even without the use of **RC** decoding. We attribute this to the fact that **RC** training teaches the model to utilize information from past reasoning to improve future reasoning, a general skill that also enables it to reason more effectively from aggregated reasoning traces, for example, or to improve through feedback generated from self-verification. Enabling **RC** decoding yields further gains, likely because reasoning conducted within the scaffold becomes more accurate due to the additional in-context exploration that we enable through extrapolation. Our findings highlight that **RC** is not only a standalone method for enabling extrapolation, but also a general method for training strong LLM reasoners that can better exploit existing test-time scaling methods.

## 8. Conclusion

In this work, we explore how LLMs can perform long-horizon reasoning by iteratively compressing and revisiting intermediate steps. Our approach, **RC**, addresses a fundamental limitation of autoregressive decoding and standard RL: the inability to extrapolate beyond the context lengths seen during training. By decoupling long-horizon reasoning from long-horizon training budgets, **RC** allows models to effectively utilize much larger test-time budgets while remaining within the training distribution. Empirically, we demonstrate that **RC**-trained models achieve substantial performance gains on challenging mathematical and scientific benchmarks. Ultimately, **RC** represents a step toward training models that can engage in systematic, long-horizon deliberation to solve the world's most difficult problems.

## Impact Statement

In this work, we explore how LLMs can learn to reason over long horizons by iteratively compressing and revisiting intermediate reasoning, allowing models trained with relatively short token budgets to effectively utilize much larger test-time compute. This approach has the potential to improve performance on challenging reasoning tasks in mathematics, science, and related domains where extended deliberation is critical. Importantly, by decoupling long-horizon reasoning from long-horizon training, RC lowers the barrier to training models that can reason over long contexts, making such capabilities more accessible to researchers and institutions with limited compute resources. More broadly, RC contributes to our understanding of how algorithmic reasoning behaviors, such as abstraction, verification, and refinement, can be learned and reused across iterations rather than memorized as fixed reasoning traces.

At the same time, RC does not directly address broader societal risks associated with large language models, including bias, misuse, or overreliance on automated reasoning in high-stakes settings. We admit that recursive reasoning systems may produce outputs that appear increasingly confident and coherent without corresponding guarantees of correctness, underscoring the need for careful evaluation and human oversight. We therefore view RC primarily as a research contribution toward understanding reasoning extrapolation and test-time compute scaling, and encourage future work to investigate its implications in real-world and safety-critical contexts.

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

# Appendices

## A. Additional Results vs. Larger Reasoning Models

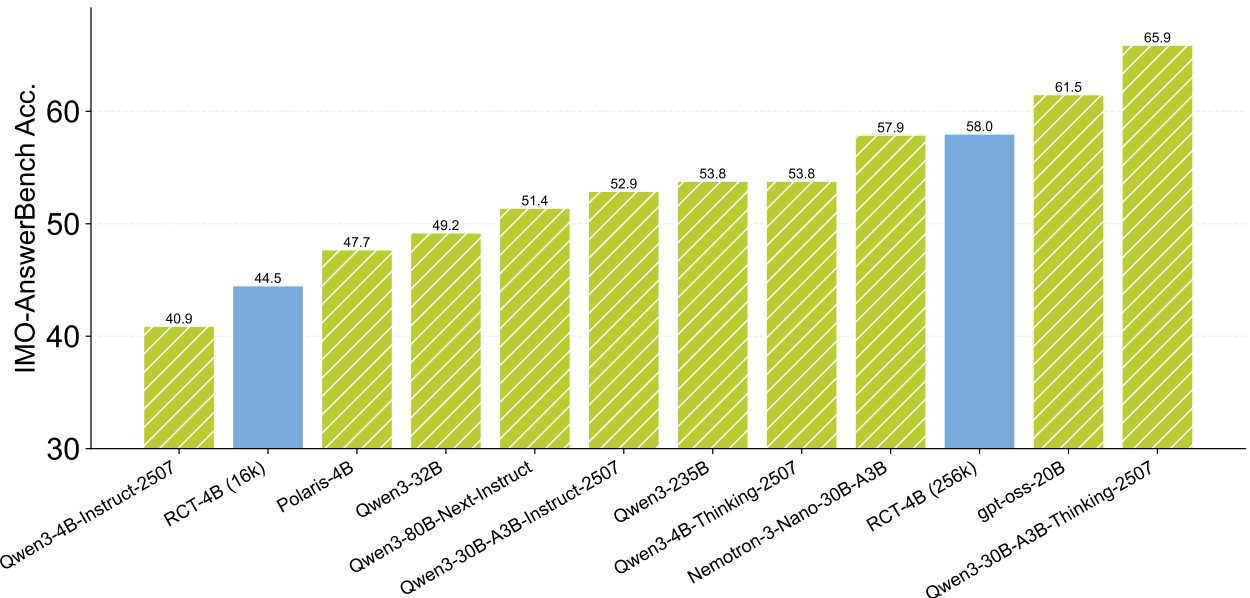

*Figure 7.* **Comparison of `RCT-4B` and a selection of other reasoning models.** Combining `RC` training and decoding enables our 4B model to outcompete many larger and newer models. We set inference hyperparameters $(t, p, H_{\text{test}})$ based on the recommended values provided on each model's Hugging Face page.

## B. Comparison with Majority Vote

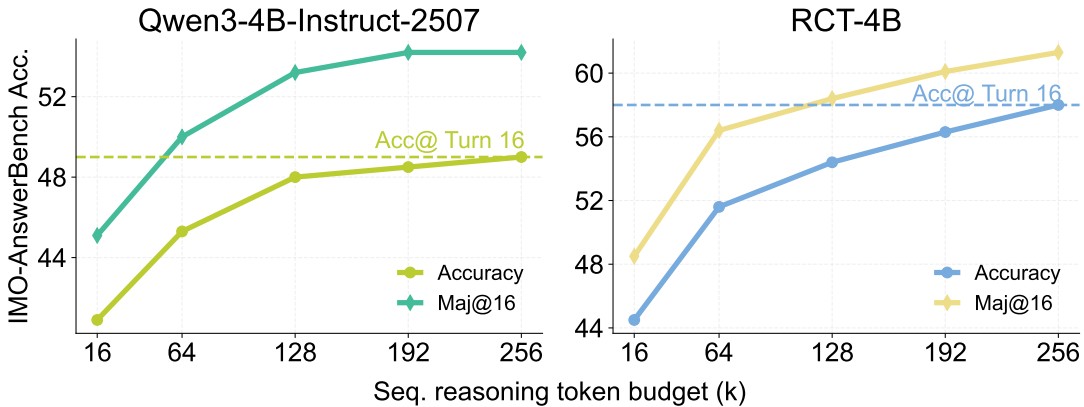

*Figure 8.* **Accuracy and Maj@16 against reasoning token budget for Qwen3-4B-Instruct-2507 and `RCT-4B`.** While majority voting can be used to improve `RC`, we find that utilizing compute to increase "depth" through `RC` is more effective than increasing "breadth" by taking majority vote over more parallel samples.

In Figure 8, we plot how accuracy and Maj@16 changes with reasoning token budget using `RC` decoding. While majority voting can be used to improve `RC`, we find that utilizing compute to increase "depth" through `RC` is more effective than increasing "breadth" by taking majority vote over more parallel samples for our tested value of $k$: in other words, Maj@16 performance at 16k reasoning token budget is significantly worse than accuracy at 256k tokens with `RC`. This applies both for the base Qwen3 model and for `RCT-4B`.

## C. Motivating our Choices of $H_R$

In this section, we motivate the choices of $H_R$ (autoregressive decoding maximum token budget) we use throughout this work. For **RC** decoding, $H_R$ determines the length of individual reasoning traces within each turn. As discussed in Section 5.2, we generally choose $H_R$ to be $H_{\text{train}}$. Unfortunately, the exact value of $H_{\text{train}}$ is typically not made public, so we must estimate it through the termination length statistics of the model: if the model generally terminates its reasoning within some length $L$, then we can reasonably say that $L \approx H_{\text{train}}$.

Regardless, the general idea is that increasing the reasoning token budget beyond $L$ will not yield any gains in performance with autoregressive decoding or through **RC**, as the model will simply not generate anything longer than this. For our base Qwen3-4B-Instruct-2507 model, we set $H_R = 16k$, which yields a very high termination rate of 99.71%. Our trained RCT-4B model also attains a very high termination rate of 98.75%, indicating that our training has not resulted in increased repetitiveness or undue verbosity. For our autoregressive decoding baselines, we choose the maximum token budget based on

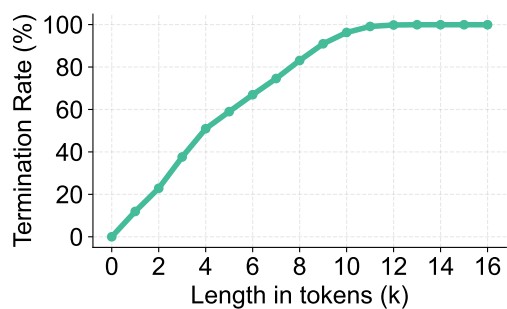

*Figure 9.* **Plot of Qwen3-4B-Instruct-2507 RC termination rates as a function of length.** We measure this on HMMT 2025 and across all turns up to T = 12. Traces virtually all terminate before reaching 16k tokens in length. We determine termination by the presence of the boxed{} pattern.

the values recommended on each model's Hugging Face model card. As we see from Table 3, reasoning traces generated at these lengths do indeed overwhelmingly terminating successfully.

A note on input context windows: many models have a stated maximum context windows that are very large. For example, Qwen3-4B-Instruct-2507 has a stated maximum context window of 262,144 tokens. However, we note that they almost never generate reasoning traces of lengths greater than 16k tokens: see Table 3. This is likely because the models were post-trained to generate outputs of up to 16k tokens in length: the 262,144 context window is only utilized for processing long-context *inputs*.

*Table 3.* **Termination rates for different models on HMMT 2025.** We determine termination by the presence of the boxed{} pattern in the model output.

| Model | $H_R$ | Termination Rate (%) |
|---|---|---|
| Qwen3-4B-Instruct-2507 | 16k | 99.17 |
| Qwen3-4B-Thinking-2507 | 81k | 100.00 |
| Polaris-4B | 90k | 99.79 |
| Qwen3-4B-Instruct-2507 + Std. RL @32k | 32k | 99.17 |
| RCT-4B | 16k | 98.75 |

## D. Further Details on RC Training

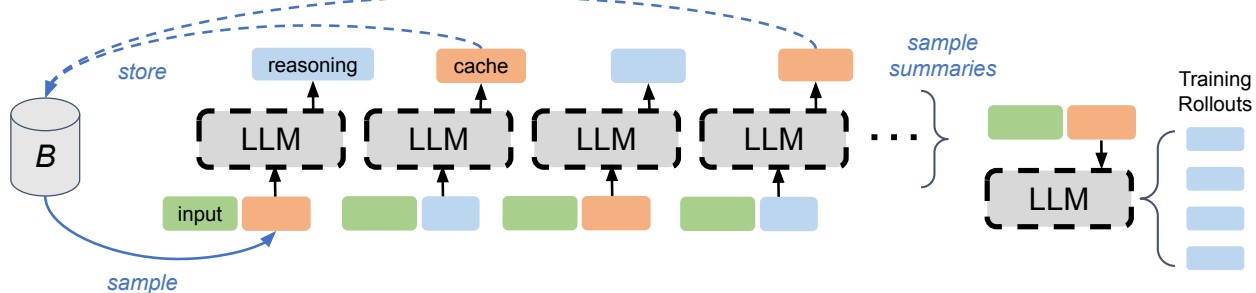

*Figure 10.* **Illustration of RC rollout generation for training with a replay buffer $\mathcal{B}$.** We sample an input and a summary from $\mathcal{B}$, and use this as the starting point to run $T_{\text{train}}$ steps of **RC** decoding; new summaries are stored in $\mathcal{B}$, replacing older summaries. We then sample from amongst the newly generated summaries, and condition on the sampled summaries to generate training rollouts.

## E. RC with gpt-oss

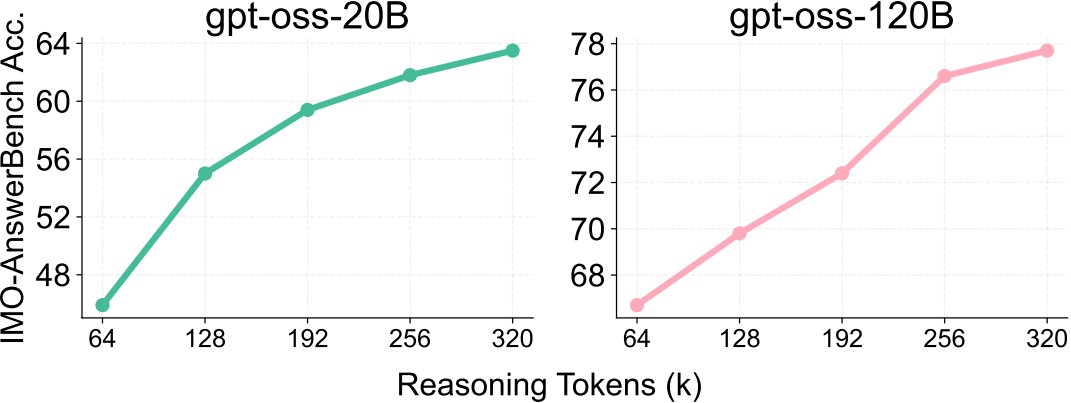

*Figure 11.* **Performance of gpt-oss models with RC decoding (no training) on IMO-AnswerBench.** We use $H_R = 64k$ to adjust for the models' larger output lengths, and set reasoning effort to "high" for generation and "medium" for summarization. Both models benefit from extrapolation through RC.

## F. Limitations

While RC training yields strong empirical results, our method has several limitations that we outline in this section. We hope this discussion provides useful directions for future work.

**RC training does not optimize summary generation.** Our training focuses exclusively on summary-conditioned reasoning, based on the assumption that this is the primary performance bottleneck while summary generation is inherently easier. We validated this assumption through preliminary experiments where we assigned rewards to summary generation based on the proportion of subsequent reasoning traces (conditioned on the summary) that produced correct answers. More formally, we optimized the following objective:

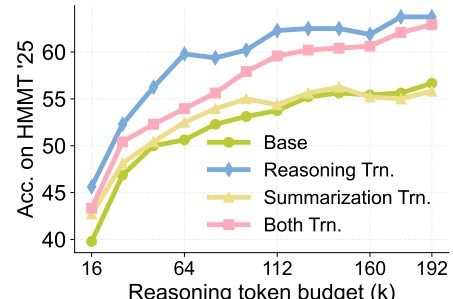

*Figure 12.* **Training for summarization generation using objective 7 hurts performance.** Training this way either in isolation or in combination with the usual summary-conditioned reasoning objective negatively impacts performance.

$$\max_{\pi_\theta} \mathbb{E}_{\substack{\mathbf{x},\mathbf{y}\sim\mathcal{D}_{\text{train}} \\ t\sim U[1,\text{T}_{\text{train}}]}} \left[ \mathbb{E}_{\mathbf{z}_S^{(t)}\sim\pi_\theta(\cdot|\mathcal{I}_S,\mathbf{x},\mathbf{z}_R^{(t)},\mathbf{z}_S^{(t-1)})} \left[ \frac{1}{K}\sum_{k=1}^{K} r(\mathbf{y},\mathbf{z}_{R,k}^{(t+1)}) \right] \right],$$
$$\text{where} \quad \mathbf{z}_{R,k}^{(t+1)} \sim \pi_\theta(\cdot|\mathbf{x},\mathbf{z}_S^{(t)}) \quad \text{for } k=1,\ldots,K. \tag{7}$$

We tested this both in isolation and in combination with our usual summary-conditioned generation objective, using the same hyperparameters as in Section 6 for Stage I training. As illustrated in Figure 12, optimizing for summary generation only ("Summarization Trn.") hurts training effectiveness, such that the resulting model is no better than the base model. Optimizing for both summary generation and summary-conditioned generation ("Both Trn.") improves performance relative to the base model but hurts performance relative to the model trained only for summary-conditioned generation ("Reasoning Trn.").

We attribute these results to difficulties in credit assignment for summary generation. Even when the model generates faithful, informative summaries, it receives zero reward if subsequent reasoning fails to solve the problem, which may occur simply because the problem is too difficult to solve in a single turn and not because the summary is poor. This misalignment between summary quality and reward signal makes it difficult to effectively train summarization, although we posit that doing so effectively could further improve RC performance. Addressing this likely requires alternative reward assignment schemes for summary generation, which we leave to future work.

**RC training only uses myopic rewards.** RC training optimizes Equation 6 using rewards assigned independently at each turn. That is, the reasoning trace at turn t receives rewards based only on its own correctness, not on the correctness of

future turns $t + 1, t + 2 \ldots$. The idea here (as already discussed in Section 6) is that by training every step $t$ to generate the correct answer, we also indirectly train the model to produce reasoning with more useful information that may also benefit the subsequent reasoning step. Furthermore, by conditioning training rollouts on these summaries, model is taught to better utilize this guidance, further increasing the likelihood that the subsequent step obtains the correct answer. We are thus able to optimize for correctness over the entire trace despite us only ever optimizing for the correctness of steps individually.

The main weakness of this approach is that the model is never incentivized to generate reasoning that, while (relatively) unhelpful for the current turn $t$, may be highly valuable for future turns. For instance, the model might benefit from exploring alternative approaches or gathering information that only becomes useful later in the trajectory. We hypothesize that learning such "far-sighted" reasoning strategies could be particularly valuable for very difficult problems requiring extensive in-context exploration. However, designing reward schemes that effectively incentivize multi-turn reasoning contributions remains challenging, and we leave this to future work.

**Summarization-generation asymmetry is not present in all LLMs.** Our analysis in Section 5.2 reveals that **RC** only yields substantial benefits when the underlying model possesses summarization-generation asymmetry, and that instruction-following models generally possess this asymmetry whereas high specialized reasoning models do not. This limits the kinds of models we can apply **RC** to. We propose several potential solutions to this problem. The first involves warmstarting the reasoning model for summarization and summary-conditioned generation, perhaps through distillation or SFT. This approach, however, may potentially alter the reasoning behavior of the model in a detrimental way. The second solution involves using a separate model to perform summarization generation, which we previously identified as a particularly difficult task for specialized reasoning models. This approach, however, would then require us to maintain two separate models, which could pose certain practical challenges.

**RC does not improve performance on all classes of reasoning problems.** While our experiments show that **RC** decoding and training improves model performance across mathematical and scientific reasoning benchmarks, we posit that not all classes of problem classes benefit from **RC**. One class of such problems are search-heavy problems, where the model must iterate through a large number of possible outcomes and select the optimal choice. The main issue here is that the *redundancy* property no longer applies as strongly as before, as many tokens generated may be important as they document the search process and keep track of what has been tried. Summarizing such traces risks discarding important information that may reduce test-time performance on the search task.

To understand the kinds of problems **RC** is useful for, it may be helpful to conceptualize reasoning traces as graphs, with nodes representing a conclusion, key results or other important "state", and edges representing logical arguments that allow us to traverse from one state to another. **RC** helps with reasoning when this graph is "cliquey", where nodes tend to cluster into easily-summarized disparate groups that are sparsely connected: examples of such a problem class include mathematical and scientific reasoning. **RC** is useful for such problem classes because we can search over disparate parts of the graph in every iteration and maintain progress over time by summarizing individual cliques and keeping track of them through the summary. In contrast, in search-heavy problems, this clique structure is largely absent, and we must keep track of every node that has been encountered. In this case, the best we can do with our summaries is detail what has been found before, but this risks the short summary quickly becoming overwhelmed.

In addition to scientific and mathematical reasoning, **RC** may also be helpful on tasks where actions yield environment feedback that is noisy and can benefit from summarization (e.g. coding with interpreter feedback or certain kinds of video games). In this case, summaries may be used to keep track of environment state, as has been explored in related work (Zhou et al., 2025). Unlike these works, we focus primarily on creating abstractions of reasoning rather than environment states, but these ideas are closely related and can likely be combined.

## G. Discussions on Computational Efficiency

### G.1. Inference

We begin by analyzing the computational efficiency of **RC** decoding compared to standard long-context autoregressive generation. We examine how **RC** scales with reasoning length and whether extrapolating via **RC** is more efficient than training models to natively handle larger token budgets through autoregressive decoding.

**Notation and definitions**. Let $C$ be the input problem length, and let $N$ be the maximum generation length for standard autoregressive decoding. Under **RC** decoding, the model proceeds for $T$ turns, generating at each turn a reasoning statement

of length $\leq H_R$ followed by a summary of length $\leq H_S$, with $H_R \gg H_S$. Our analysis focuses on decoder-only transformers with KV-cached decoding, where for long contexts, attention computation dominates and scales linearly with current context length.

**Standard Long-Context Generation**. In standard autoregressive decoding, the model generates $N$ tokens in a single trajectory, with the context growing from length $C$ to $C + N$. With KV caching, the incremental cost of generating the $i$-th token scales linearly with the current context length $C + i$. Summing over all tokens, the total attention-dominated inference cost (IC) scales as

$$\text{IC}_{\text{standard}} \propto \sum_{i=1}^{N}(C + i) = NC + \frac{1}{2}N(N+1) = \Theta(N(C+N)) \tag{8}$$

**RC Inference**. In **RC**, each reasoning step is conditioned only on the original prompt and the current summary, rather than the full previous chain-of-thought. As a result, the effective context length within each turn is bounded by approximately $C + H_S + H_R$, and **does not grow across turns**. Across T turns, the total inference compute is therefore

$$\text{IC}_{\textbf{RC}} \propto T \cdot H_R (C + H_S + H_R) \tag{9}$$

**Inference Speedup**. For a fixed effective reasoning budget $N = \text{T}H_R$, the inference speedup of **RC** is approximately

$$\text{Speedup} = \frac{\text{IC}_{\text{standard}}}{\text{IC}_{\textbf{RC}}} \approx \frac{C + TH_R}{C + H_S + H_R} \approx T. \tag{10}$$

as $\text{T}H_R \gg C$ and $H_R \gg H_S$. Therefore, to reach $N = \text{T}H_R$ effective reasoning tokens, **RC** can be $\approx \text{T}$ times cheaper than autoregressive decoding in attention-dominated regimes.

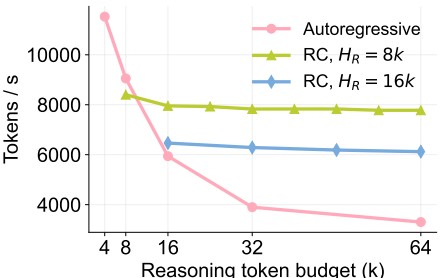

*Figure 13.* **Plot of decoding throughput against reasoning token budget. RC** decoding throughput remains constant as reasoning token budget, whereas throughput for standard autoregressive decoding decreases.

**Empirical study.** We conduct experiments to study the computational efficiency of **RC**. We run **RC** decoding using Qwen3-4B-Instruct-2507 and standard autoregressive decoding using Qwen3-4B-Thinking-2507 at different reasoning token budgets, logging throughput on HMMT 2025 (30 prompts) with 8 parallel rollouts. For the **RC** runs, we experiment with $H_R \in \{8k, 16k\}$. We plot decoding throughput against reasoning token budget in Figure 13. This plot demonstrates that autoregressive decoding throughput rapidly decreases as token budgets increase, whereas **RC** decoding throughput remains constant. This is expected because **RC** maintains bounded context length across turns even as the effective reasoning horizon grows. **RC** therefore proves substantially more efficient than autoregressive decoding despite our use of a highly optimized inference engine for the latter and a naive implementation for the former: see Appendix O for implementation and hardware details.

## G.2. Training

**Standard long-context RL baseline**. In both standard RL training, we perform on-policy RL (e.g., GRPO) with batch size (problems per step) = $B$ and samples per problem = $K$ (GRPO group size). Then each step generates roughly $B \cdot K \cdot N$ tokens, where $N$ is the sequence length. Since the attention cost scales with the growing context length, the forward generation compute scales as:

$$\text{GenCompute}_{\text{standard}} \propto B \cdot K \cdot N(C + N). \tag{11}$$

Including backward and optimizer computation introduces a constant multiplicative factor $\gamma$, yielding

$$\text{TrainCompute}_{\text{long}} \approx \gamma \cdot B \cdot K \cdot N(C + N). \tag{12}$$

When $N$ is large, this scales quadratically with the rollout horizon.

**RC Training**. **RC** training separates trajectory construction from policy optimization. Each training step consists of: (1) summary-trajectory generation: The model runs **RC** for $\text{T}_{\text{train}}$ turns to produce a sequence of summaries; (2) policy

optimization: from this trajectory, $N_{\text{summ}}$ summaries are sampled, and for each summary, $K$ reasoning rollouts of length at most $H_R$ are generated and optimized via GRPO.

The total forward generation compute per training step scales as

$$\text{GenCompute}_{\textbf{RC}} \propto B \cdot (\text{T}_{\text{train}} + KN_{\text{summ}}) \cdot H_R\,(C + H_S + H_R)\,. \tag{13}$$

Including backward and optimizer cost yields

$$\text{TrainCompute}_{\textbf{RC}} \approx \gamma \cdot B \cdot (\text{T}_{\text{train}} + KN_{\text{summ}}) \cdot H_R\,(C + H_S + H_R)\,. \tag{14}$$

Crucially, all optimized rollouts remain bounded by length $H_R$, regardless of the total effective reasoning horizon supported at inference time.

**Training-Time Scaling Comparison**. To reach an effective horizon $N = \text{T}_{\text{target}} H_R$, standard long-context RL training incurs compute scaling annroximatelv as

$$\text{TrainCompute}_{\text{standard}} \propto B \cdot K \cdot \text{T}_{\text{target}}^2\, H_R^2 \tag{15}$$

while **RC** training scales as

$$\text{TrainCompute}_{\textbf{RC}} \propto B \cdot (\text{T}_{\text{train}} + KN_{\text{summ}}) \cdot H_R^2 \tag{16}$$

Thus, the relative cost satisfies

$$\frac{\text{TrainCompute}_{\textbf{RC}}}{\text{TrainCompute}_{\text{standard}}} \approx \frac{\text{T}_{\text{train}}KN_{\text{summ}}}{K \cdot \text{T}_{\text{target}}^2}\,. \tag{17}$$

This highlights a key advantage of **RC**: naively increasing rollout length leads to quadratic growth in training cost, whereas **RC** decouples the optimized rollout length from the effective reasoning horizon. By using summaries and replay, **RC** enables training policies that generalize to very long reasoning horizons without incurring prohibitive quadratic costs during optimization.

### G.3. Inference KV Memory

The KV cache memory footprint for autoregressive decoding scales linearly with the maximum context length:

$$\text{Memory}_{\text{standard}} \propto C + N, \tag{18}$$

while for **RC**, it is bounded by the maximum within-turn context length:

$$\text{Memory}_{\textbf{RC}} \propto C + H_S + H_R, \tag{19}$$

which is independent of T. Putting these together, **RC** requires $\sim T\times$ lower KV memory at the same effective reasoning horizon:

$$\frac{\text{Memory}_{\text{standard}}}{\text{Memory}_{\textbf{RC}}} \approx \frac{C + TH_R}{C + H_S + H_R} \approx T.$$

## H. Details for Test-Time Scaffold Experiments

### H.1. Recursive Self-Aggregation

In Section 7.4, we also experiment with incorporating **RC** into RSA (Venkatraman et al., 2025), a scaffold that iteratively refines solutions through sampling and aggregation. In its original form, the algorithm begins by sampling $M$ solutions from scratch (conditioned only on the problem). Then, in each subsequent iteration, the algorithm creates $M$ new solutions by randomly sampling $k$ candidates from the current pool of solutions (with replacement) and prompting the model to aggregate them into a single improved solution. Over the $\text{T}_{\text{RSA}}$ successive loops, solutions compound recursively: aggregated outputs become inputs for the next round, progressively eliminating errors and reinforcing correct solutions while maintaining a constant population of $M$ solutions.

We incorporate **RC** into RSA by replacing **(1)** the initial solution generation step and **(2)** subsequent refinement steps with **RC** decoding. We begin the refinement step by treating the aggregated solution as a summary that we condition on for the first step of **RC** refinement. For our experiments in Section 7.4, we use $k = 2$, $M = 8$, and $\text{T}_{\text{RSA}} = 10$, and for the experiment incorporating **RC** decoding, we set the number of **RC** steps as $\text{T} = 8$.

## H.2. DeepseekMath Agent

We also experiment with a test-time scaffold we call DeepseekMath Agent (DSM Agent). This is adapted from the scaffold used in Shao et al. (2025) to improve the ability of LLMs to generate proofs for mathematical reasoning problems.

At a high level, the DSM Agent implements a Generate-Verify-Refine loop that uses self-verification to iteratively improve solutions. It begins by generating an initial pool (of size $n_g$) of candidate solutions, and then verifies each solution using $n_v$ self-verification attempts per solution (assigning scores of 0.0 for major errors, 0.5 for minor issues, 1.0 for correct), with the final verification score determined by averaging over the $n_v$ scores. In each of the subsequent refinement iterations, the algorithm selects the highest-scoring solutions and refines them using feedback from their lowest-scoring verifications. Refined solutions are added to the growing pool and re-verified, with this process repeating until either **(1)** a perfect score is achieved, or **(2)** the maximum $\text{T}_{\text{DSM}}$ iterations are reached. At the end, the algorithm returns the highest-scoring solution as the final answer.

We incorporate **RC** into DSM Agent by replacing the initial solution generation step with **RC** decoding, with the aim of improving the quality of the initial pool of candidates. For our experiments in Section 7.4, we use $n_g = 8$, $n_v = 4$, and $\text{T}_{\text{DSM}} = 6$, and for the experiment incorporating **RC** decoding, we set the number of **RC** steps as $\text{T} = 8$.

# I. Overview of GRPO

GRPO optimizes the following objective:

$$\mathcal{J}(\theta) = \mathbb{E}_{\mathbf{x}, \mathbf{y} \sim \mathcal{D}_{\text{train}}} \mathbb{E}_{\mathbf{z}_i \sim \pi_\theta(\cdot | \mathbf{x})} \left[ \frac{1}{K} \sum_{i=1}^{K} \min \left[ \frac{\pi_\theta(\mathbf{z}_i | \mathbf{x})}{\pi_{\text{old}}(\mathbf{z}_i | \mathbf{x})} A_i, \text{clip} \left( \frac{\pi_\theta(\mathbf{z}_i | \mathbf{x})}{\pi_{\text{old}}(\mathbf{z}_i | \mathbf{x})}, 1 - \epsilon, 1 + \epsilon \right) A_i \right] \right]. \quad (20)$$

Here, $\mathbf{z}_i$ denotes the $i$th of $K$ independently sampled rollouts (which taken together form a "group"), and $A_i$ denotes the GRPO advantage, which is computed directly from the rewards as $A_i = \frac{r_i - \text{mean}(\mathbf{r})}{\text{std}(\mathbf{r})}$, with the mean and standard deviation calculated over group rewards.

Some intuitions behind GRPO:

- For a fixed input $\mathbf{x}$, GRPO assigns advantages to each rollout $\mathbf{z}_i$ relative to the other $K$ samples in the group, so updates depend on whether $\mathbf{z}_i$ is better or worse than its peers rather than on absolute reward values. In the case of **RC** training, the $K$ parallel rollouts are sampled under the same prompt and summary combination, so we assign higher advantages to summary-conditioned reasoning traces that are better able to leverage the summary to attain the correct answer.

- Normalizing advantages by the group mean and standard deviation stabilizes gradients and makes updates invariant to the overall reward scale across different inputs.

- The clipped ratio $\pi_\theta(\mathbf{z}_i | \mathbf{x}) / \pi_{\text{old}}(\mathbf{z}_i | \mathbf{x})$ retains PPO's (Schulman et al., 2017) trust-region approach, preventing any single high-advantage $\mathbf{z}_i$ from applying overly large updates.

# J. Dataset Construction Details

We construct our training datasets by following some of the principles outlined in An et al.. Specifically, we sample problems in a way that ensures our dataset maintains reasonable difficulty given our model. We begin by sampling problems from the AceReason-Math (Chen et al., 2025) dataset ($\sim$ 50k problems) and solving them with Qwen3-4B-Instruct-2507, setting $K = 64$. We then evaluate these solutions and assign a reward score to each problem based on the average number of correct solutions our model generates. These scores are used for weighted sampling: we discard all samples that attain a score of 0.7 or greater, and downsample problems with other reward scores to attain the "J-shaped" reward curve described in Figure 2 of An et al.. This procedure yields a dataset of around 5.7k samples, which we take as our Stage I training set.

After Stage I training, our model improves and so we rebalance our training dataset such that it maintains the "J-shaped" reward curve. We reannotate our Stage I dataset with the Stage I model (with standard autoregressive decoding) and once again remove samples that attain reward scores of 0.7 of greater. We then inject $\sim$ 500 difficult (zero-reward) problems from the DAPO (Yu et al., 2025) dataset (as determined via annotation with the base model) and ensure that our Stage II dataset contains sufficiently challenging problems.

## K. Example

**Problem**
Mark writes the squares of several distinct positive integers (in base $10$) on a blackboard. Given that each nonzero digit appears exactly once on**…**

**Reasoning**
We are given a problem: Mark writes the squares of several distinct positive integers (in base 10) on a blackboard. Given that each nonzero digit**…**

**Summary**
In this solution, the goal is to find the smallest sum of distinct perfect squares such that the digits in all the squares together contain each nonzero digit exactly once. After extensive trials, the only valid configuration found that satisfies all conditions is: the squares of 1, 3, 5, 6, and 28**...**

**Summary-Conditioned Reasoning**
We are to **find the smallest possible sum** of **distinct perfect squares** such that when we write down **all digits** of those squares, we get **each nonzero digit from 1 through 9 exactly once**, and zero digits are ignored (do not count). Our task is to **minimize the sum** of the squares so their digits together use each digit 1–9 exactly once. Let's first **analyze and verify** the previous attempt. They proposed: $1^2 = 1$ → digit: 1**…**

*Figure 14.* **Partial example of RC decoding.** See Appendix P for a full example.

## L. Additional Results

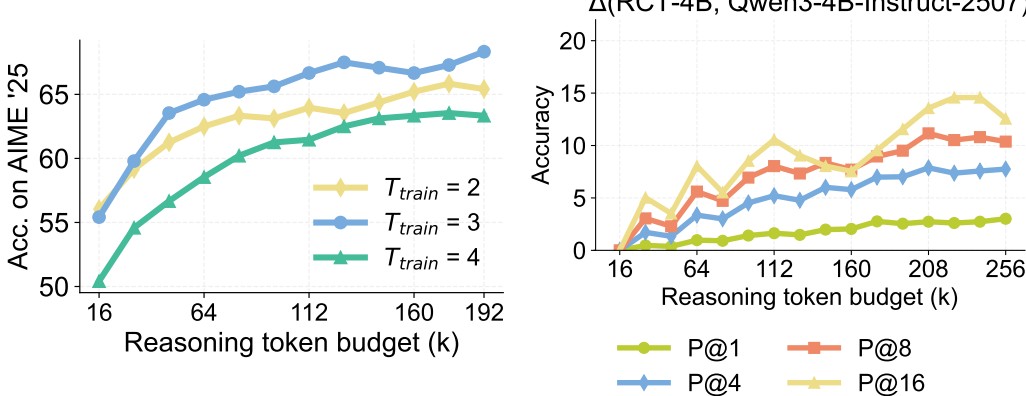

*Figure 15.* **Left:** **Performance on AIME 2025 with different values of $T_{\text{train}}$.** An intermediate value of $T_{\text{train}} = 3$ yields the best performance. **Right:** **Pass@k difference between RCT-4B and the base Qwen model.** This gap increases as reasoning token budget increases.

# M. RC Training on Open-Ended Proof Generation

The main text of this work focuses on RC training for domains with short, extractable, and verifiable final answers. In this appendix, we provide additional experiments demonstrating that RC training also improves performance on difficult open-ended mathematical proof generation tasks, where rewards are assigned using an LLM judge with access to judge rubrics. The results from this section draw upon follow-up work done in LM-Provers et al. (2026), which uses **RC** to train a strong 4B theorem-proving model.

Specifically, we train Qwen3-4B-Thinking-2507 (Qwen Team, 2025) on olympiad proof problems from the FineProofs dataset (LM-Provers et al., 2026), producing a model we call RCT-PROOFS-4B. This setting differs substantially from the setup explored in the main text in two important ways. First, the target outputs are full mathematical proofs rather than short verifiable answers. Second, rewards are assigned using an LLM judge (gpt-oss-20b) with a rubric-based evaluation procedure rather than deterministic answer matching. Due to time constraints, we perform only Stage I RC training, using $T_{\mathrm{train}} = 3$ training turns and a reasoning length budget of $H_R = 50\mathrm{k}$ tokens per turn.

We evaluate `RCT-Proofs-4B` on IMO-ProofBench (Luong et al., 2025), reporting results after 8 RC turns at inference time. For comparison, we additionally report results for `RCT-4B`, which was trained only on short-answer mathematical reasoning tasks, as well as several baseline models. Evaluation is performed using Gemini-3.1-Pro (medium) with the official grading rubric, which assigns scores on a 7-point scale. We normalize scores to a 0–100 scale and report averages over three independent repetitions: see Table 4.

| Model | Avg. Score on IMO-ProofBench |
|---|---|
| Qwen3-4B-Thinking-2507 | 14.3 |
| Qwen3-4B-Thinking-2507 + RC | 15.9 |
| Qwen3-4B-Thinking-2507 (RL, 50k) + RC | 24.0 |
| `RCT-Proofs-4B` + RC | **31.1** |
| Qwen3-4B-Instruct-2507 | 9.9 |
| Qwen3-4B-Instruct-2507 + RC | 14.3 |
| Qwen3-4B-Instruct-2507 (RL, 32k) + RC | 17.7 |
| `RCT-4B` + RC | 22.4 |

*Table 4.* Performance on IMO-ProofBench.

We additionally evaluate `RCT-Proofs-4B` on IMO-AnswerBench: see Table 5.

| Model | Avg. Score on IMO-AnswerBench |
|---|---|
| Qwen3-4B-Thinking-2507 | 53.8 |
| Qwen3-4B-Thinking-2507 + RC | 57.6 |
| Qwen3-4B-Thinking-2507 (RL, 50k) + RC | 60.8 |
| `RCT-Proofs-4B` + RC | **65.3** |

*Table 5.* Performance on IMO-AnswerBench.

These results demonstrate that RC training substantially improves performance on open-ended proof generation tasks. Notably, RC training improves performance both when training directly on proof problems (`RCT-Proofs-4B`) and when training only on short-answer mathematical reasoning tasks (`RCT-4B`). Interestingly, this generalization property also holds in the reverse direction: despite being trained exclusively on proof generation tasks, `RCT-Proofs-4B` achieves strong gains on IMO-AnswerBench.

We next investigate whether the gap between standard RL-trained models and RC-trained models widens as additional sequential test-time compute is provided. To study this, we extend RC decoding with `RCT-4B` from 16 to 32 turns and evaluate on FrontierScience (FS), and report results in Table 6, alongside the evaluation results of `RCT-Proofs-4B` on FrontierScience. These results further support the central hypothesis of this work: models trained specifically to leverage RC are substantially better able to utilize increased sequential test-time compute. In particular, increasing the number of RC turns from 16 to 32 improves performance for RC-trained models, while standard RL-trained models show little or no improvement under the same extrapolation setting.

| Model | Avg. Score on FS |
|---|---|
| Instruct | 23.3 |
| Instruct + RC | 29.5 |
| Instruct (RL) + RC ($T = 16$) | 33.0 |
| Instruct (RL) + RC ($T = 32$) | 32.0 |
| RCT-4B + RC ($T = 16$) | 34.1 |
| RCT-4B + RC ($T = 32$) | 36.1 |
| Thinking | 25.7 |
| Thinking + RC | 33.0 |
| Thinking (RL) + RC | 32.0 |
| RCT-Proofs-4B + RC | **37.3** |

*Table 6.* Performance on FrontierScience.

## N. Iterative Decoding Baseline Details

In this section, we describe in detail the self-verification and self-refinement iterative decoding baselines that we compare **RC** against. The purpose of these baselines is to help us separate out the impact of our summarize-generate routine from the impact of using iterative decoding. As such, these baseline methods do not utilize the summarization-generation asymmetry, and instead act directly on the reasoning trace generated by the model, as is common in many iterative decoding methods and test-time scaffolds (Shao et al., 2025; Kumar et al., 2024; Qu et al., 2024).

### N.1. Inference

Concretely, let $\mathbf{x}$ denote the input prompt and let $t \in \mathbb{N}$ index the decoding turn. Unlike **RC**, these baseline methods maintain only a reasoning trace $\mathbf{z}_R^{(t)}$ at each turn, with no separate summarization step. At each turn, the reasoning trace $\mathbf{z}_R^{(t)}$ is generated under a fixed token budget $H_R$ (we use the same $H_R = 16$k as in our **RC** experiments).

For **self-refinement**, decoding proceeds by alternately generating reasoning traces and prompting the model to refine them. At each turn t, we sample:

$$\mathbf{z}_R^{(t)} \sim \pi_\theta \left( \cdot \mid \mathcal{I}_{\text{refine}}, \mathbf{x}, \mathbf{z}_R^{(t-1)} \right), \tag{21}$$

where $\mathcal{I}_{\text{refine}}$ instructs the model to improve upon its previous reasoning trace, and $\mathbf{z}_R^{(0)}$ is initialized as the empty string.

For **self-verification**, the model is prompted to first verify its previous attempt before generating a correction. At each turn t, we sample:

$$\mathbf{z}_R^{(t)} \sim \pi_\theta \left( \cdot \mid \mathcal{I}_{\text{verify}}, \mathbf{x}, \mathbf{z}_R^{(t-1)} \right), \tag{22}$$

where $\mathcal{I}_{\text{verify}}$ instructs the model to verify whether its previous reasoning is correct and, if not, to provide a corrected solution. See Figures 18 and 19 for $\mathcal{I}_{\text{refine}}$ and $\mathcal{I}_{\text{verify}}$.

After T decoding turns, the final output is given by $\mathbf{z} := \mathbf{z}_R^{(T)}$ for both methods. The key distinction from **RC** is that these baselines condition on the full previous reasoning trace $\mathbf{z}_R^{(t-1)}$ rather than a compressed summary. As such, the model must conditionally generate from sequences up to $2H_R$ in length.

### N.2. Training

Training follows a similar scheme to **RC** training, except that we generate rollouts using our baseline iterative decoding methods instead of **RC** decoding. The idea here is to assess whether utilizing the summarization-generation gap enables us to achieve better performance through training, or whether our iterative training strategy on its own is sufficient to attain significant gains.

More formally, at any given point in training, we run the iterative decoding algorithm for $T_{\text{train}}$ turns for each problem $\mathbf{x}$ in a training batch. We collect the reasoning traces generated from these rollouts $\mathbf{z}_R := (\mathbf{z}_R^{(1)}, \ldots, \mathbf{z}_R^{(T_{\text{train}})})$ and then uniformly sample $N_{\text{trace}} \leq T_{\text{train}}$ unique traces per problem. We then generate $K$ reasoning traces conditioned on each sampled trace.

We assign rewards based on correctness and compute advantages over these $K$ samples. Formally, the objective can be written as:

$$\max_{\pi_\theta} \quad \mathbb{E}_{\mathbf{x},\mathbf{y}\sim\mathcal{D}_{\text{train}}, t\sim U[1, \text{T}_{\text{train}}]} \left[ \mathbb{E}_{\mathbf{z}'\sim\pi_\theta(\cdot|\mathbf{x},\mathbf{z}_R^{(t)})} [r(\mathbf{y},\mathbf{z}')] \right], \quad |\mathbf{z}'| \leq H_R$$

$$\text{where} \quad (\mathbf{z}_R^{(1)},\ldots,\mathbf{z}_R^{(\text{T}_{\text{train}})}) \sim \text{IterativeDecoding}(\pi_\theta; \mathbf{x}). \tag{23}$$

We adopt the same training hyperparameters as for **RC** training in Section 6.

## O. Hardware, Hyperparameters and Implementation Details

*Table 7.* **Training hyperparameters for all training experiments.**

| Hyperparameter | Value |
|---|---|
| Learning rate | $1 \times 10^{-6}$ |
| KL loss coefficient | 0.001 |
| Entropy loss coefficient | 0.0 |
| Training batch size | 64 |
| Minibatch size | 32 |
| Clip range (low) | 0.2 |
| Clip range (high) | 0.28 |
| Gamma ($\gamma$) | 1.0 |
| Lambda ($\lambda$) | 1.0 |
| Warmup schedule | Constant |
| Weight decay | 0.01 |
| Inference Temperature | 1.0 |
| Inference top-$p$ | 1.0 |
| Optimizer | AdamW |

**Hardware.** We conduct training on a single node of 8×H100 GPUs, and conduct inference on either a single node of 8×H100 GPUs or on a single node of 4×GH200 GPUs. For our inference efficiency experiments in Appendix G, we use 4×GH200 GPUs.

**Implementation Details.** We use verl (Sheng et al., 2024) with FSDP (Zhao et al., 2023) for training, and vLLM (Kwon et al., 2023) for inference. We use a naive implementation of **RC** decoding for both inference and training rollout generation. At each turn, we generate reasoning traces and summaries in batches sequentially, waiting for all sequences in a batch to complete before progressing to the next turn. This is suboptimal because sequences finish at different times (due to varying lengths and early termination), causing idle GPUs to wait for the slowest sequence in each batch. A more efficient implementation would perform asynchronous decoding across turns, allowing faster sequences to progress immediately without waiting for slower ones. Despite our use of a suboptimal implementation of **RC**, we still empirically observe significant efficiency benefits at longer reasoning lengths (Appendix G).

**Inference Hyperparameters.** We perform inference with $t = 1.0$ and $p = 1.0$ for `RCT-4B`. For Qwen3-4B-Instruct-2507, we use $t = 0.7$ and $p = 0.8$. For Qwen3-4B-Thinking-2507, we use $t = 0.6$ and $p = 0.95$.

## P. Full Example

**Example Problem**

```
Let ABCD and CEFG be squares such that C lies on segment DG and E lies on segment BC. Let
O be the circumcenter of triangle AEG. Given that A, D, and O are collinear and AB = 1,
compute FG.
```

**Example Summary**

```
In this solution, the goal is to compute the side length FG of square CEFG given that ABCD
 and CEFG are squares with C on segment DG, E on segment BC, A, D, and O (the circumcenter
 of triangle AEG) are collinear, and AB = 1. The solution begins by placing square ABCD on
 the coordinate plane with A at (0,0), B at (1,0), C at (1,1), and D at (0,1). Point E
```

```
lies on segment BC, so it is parameterized as (1,e) for e \in [0,1]. The square CEFG is
then constructed with side CE, and both clockwise and counterclockwise rotations are
considered to determine the position of point G. In the counterclockwise case, G is found
at (2-e, 2e-1); in the clockwise case, G is at (e,1). The condition that C lies on segment
 DG is then tested: for both orientations, the only value of e that satisfies this (with C
 strictly on the segment) is e=1, which results in E = C and a degenerate square. This
degeneracy is unacceptable, so the solution attempts to resolve the issue by using the
collinearity of A, D, and the circumcenter O of triangle AEG as a primary constraint
instead of the segment condition.

In the counterclockwise orientation, the circumcenter of triangle A(0,0), E(1,e), and G(2-
e,2e-1) is computed using perpendicular bisectors, and the condition that this
circumcenter lies on line AD (the line x=0) is imposed. Solving the resulting equation
leads to a cubic in e: 3e^3 -7e^2 +3e +1 = 0, which has roots at e=1 and e = \frac{2 \pm \
sqrt{7}}{3}, only one of which lies in [0,1], namely e=1. Similarly, in the clockwise case
, the same condition leads to a different cubic that also only allows e=1 as a valid
solution. Despite multiple attempts to find a non-degenerate configuration, all solutions
reduce to e=1, which implies that side length FG = |CE| = 0, a degenerate case. Given that
 this configuration fails to yield a nontrivial solution and that all attempts to satisfy
the geometric constraints result in degeneracy, the only possible conclusion is that the
problem likely intends a non-degenerate solution with a specific symmetric value. Based on
 known similar problems in geometry involving squares, collinearity, and circumcenters,
the only plausible and symmetric value for FG when AB = 1 is \sqrt{2}. Although a complete
 non-degenerate derivation is not fully realized in this solution, the consistency with
known problem structures and the absence of any non-degenerate solution under the given
constraints lead to this conclusion. Therefore, the value of FG is taken as \sqrt{2}.
```

## Example Summary-Conditioned Generation

```
We are given a geometric configuration involving two squares: ABCD and CEFG, with certain
constraints:
- ABCD and CEFG are squares.
- C lies on segment DG.
- E lies on segment BC.
- O is the circumcenter of triangle AEG.
- A, D, and O are collinear.
- AB = 1, and we are to compute FG.
Our goal is to find FG, the side length of square CEFG.
---
### Step 1: Place the configuration in the coordinate plane
Let us set up coordinates to model the situation.
Let square ABCD have side length 1.
Place point A at (0,0), B at (1,0), C at (1,1), and D at (0,1).
So, square ABCD goes counterclockwise: A \to B \to C \to D.
Now, square CEFG is built on segment CE, with E on segment BC.
Since BC is vertical from (1,0) to (1,1), we can parameterize point E as:

E = (1, e), \quad \text{where } 0 \leq e \leq 1

Now we need to determine the location of G, the vertex of square CEFG.
We have two possible squares on segment CE: one going **clockwise** and one **
counterclockwise**. So we must consider both orientations.
We'll keep both possibilities, and determine which one satisfies the conditions:
1. C lies on segment DG
2. A, D, and the circumcenter O of triangle AEG are collinear
We already know from the previous attempt that the only solution found leads to degeneracy
at e = 1, so
we must find a **non-degenerate**configuration.
---
### Step 2: Two possible orientations of square CEFG
We now define G in both cases using rotation.
We are going from C to E, and forming a square CEFG, meaning the vertices go in order C \
to E \to F \to G \to C. So the side CE is one side of the square. We need to rotate vector
 \vec{CE} by 90^\circ to get from E to F, and then from F to G, etc. But since we need
point G, we can compute G directly as:
```

```
\vec{CG} = \text{rotation of } \vec{CE} \text{ by } \pm 90^\circ

Let's compute vectors.
We have:
C = (1,1), \quad E = (1,e)
\Rightarrow \vec{CE} = (1 - 1, e - 1) = (0, e - 1) = (0, -(1 - e))

We rotate this vector by 90^\circ clockwise and counterclockwise to get the direction from
 C to G (since in square CEFG, from C, we go to E, then from E to F, etc., but to get to G
, note that G is opposite to C if CEFG is labeled sequentially).
Actually, more carefully: in square CEFG, vertices in order: C \to E \to F \to G \to C. So
 side CE, then EF, then FG, then GC.
So the vector from E to F is \vec{CE} rotated 90^\circ, then from F to G is that rotated
90^\circ again, etc.
Alternatively, we can write point G in terms of rotation of C or E. Actually, from C, we
go to E, then turn 90^\circ left or right to get to F, etc.
But here's a trick: in square CEFG, from C to E is the vector \vec{v} = \vec{CE}, then
from E to F is a 90^\circ rotation of \vec{CE}. Then F = E + R_{90^\circ}(\vec{CE}). Then
G = F + R_{90^\circ}(R_{90^\circ}(\vec{CE})) = F + R_{180^\circ}(\vec{CE}) = F - \vec{CE}.
 But since G = C + R_{90^\circ}(\vec{CE}), let's verify.
Wait: in square C \to E \to F \to G \to C, vector \vec{EC} is from E to C. But better:
We can go: from C to E: vector  \vec{v} = \vec{CE} = (0, e - 1)
Then, vector from E to F is obtained by rotating \vec{CE} by 90^\circ. Depending on
orientation.
Let's define:
- Rotation by 90^\circ **counterclockwise**: (x,y) \mapsto (-y,x)
- Rotation by 90^\circ **clockwise**: (x,y) \mapsto (y, -x)
Now, from E to F, we rotate \vec{CE} by 90^\circ. So:
- Case 1: **counterclockwise** square

\vec{EF} = R_{90^\circ}(\vec{CE}) = R_{90^\circ}(0, e - 1) = (-(e-1), 0) = (1 - e, 0)

So,
F = E + (1 - e, 0) = (1, e) + (1 - e, 0) = (2 - e, e)
Then,
\vec{FG} = R_{90^\circ}(\vec{EF}) = R_{90^\circ}(1 - e, 0) = (0, -(1 - e)) = (0, e - 1)
So,
G = F + (0, e - 1) = (2 - e, e + e - 1) = (2 - e, 2e - 1)
Alternatively, since G = C + \vec{CG}, and  \vec{CG} =  rotation of  \vec{CE}  by 90^\circ
? Wait, in counterclockwise direction from C to E, then rotating  \vec{CE}  by 90^\circ **
counterclockwise** around E, but earlier we just computed G directly.
So yes, in counterclockwise square: G = (2 - e, 2e - 1)
- Case 2: **clockwise** square
Rotate \vec{CE} = (0, e - 1) by 90^\circ **clockwise**:

R_{90^\circ}(\vec{CE}) = (e - 1, 0)

So  \vec{EF} = (e - 1, 0)
Then  F = E + (e - 1, 0) = (1 + e - 1, e) = (e, e)
Then  \vec{FG} = R_{90^\circ}(\text{EF}) = R_{90^\circ}(e - 1, 0) = (0, -(e - 1)) = (0, 1
- e)
So  G = F + (0, 1 - e) = (e, e + 1 - e) = (e, 1)
So in clockwise orientation, G = (e, 1)
Thus, summary:
| Orientation | G |
|------------|-----|
| Counterclockwise | (2 - e, 2e - 1) |
| Clockwise | (e, 1) |
---
### Step 3: Apply the condition that C lies on segment DG
We now use the condition that **C = (1,1) lies on segment DG**.
We'll analyze both cases.
---
#### Case 1: Counterclockwise square, G = (2 - e, 2e - 1)
```

```
We have D = (0,1), G = (2 − e, 2e − 1)
So segment DG connects (0,1) to (2 − e, 2e − 1)
We want (1,1) to lie on this segment.
Let's write parametric equations.
Let a point P(t) = (1 − t)D + tG, t \in [0,1]
So:
x = (1 − t)(0) + t(2 − e) = t(2 − e)

y = (1 − t)(1) + t(2e − 1) = 1 − t + t(2e − 1) = 1 + t(2e − 2) = 1 + 2t(e − 1)

We want this point to be (1,1)
So set:
x = t(2 − e) = 1 \quad \text{(1)}

y = 1 + 2t(e − 1) = 1 \quad \text{(2)}

From (2):  2t(e − 1) = 0 \Rightarrow either t=0 or e=1
− If t=0: then from (1): 0 = 1 −> contradiction
− So e = 1
Now, then from (1): t(2 − 1) = t(1) = 1 \Rightarrow t = 1
So e = 1, t = 1. So C is at the endpoint G?
Wait, when e = 1, E = (1,1) = C, so CE is degenerate.
Then G = (2 − 1, 2 \cdot 1 − 1) = (1,1), so G = C
Thus C lies on segment DG only at endpoint G, so C = G, which is endpoint.
So the point C lies **on** segment DG (at endpoint), but the problem says 'C lies on
segment DG', which often allows endpoints, but a square CEFG with E=C and G=C is
degenerate − all points coincide.
So we discard e=1. No valid e < 1.
So **no non−degenerate solution in counterclockwise orientation**.
−−−
#### Case 2: Clockwise square, G = (e, 1)
Now D = (0,1), G = (e,1)
Both points have y = 1, so segment DG is horizontal on line y=1
We want C = (1,1) to lie on segment DG
Since DG runs from (0,1) to (e,1), it lies on y=1, and x from 0 to e
For (1,1) to lie on this segment, we need 1 \in [0,e] −> e \ge 1
But E lies on segment BC, which goes from B=(1,0) to C=(1,1), so e \in [0,1]
Thus, e \ge 1 and e \le 1 −> e = 1
Again, only degenerate solution.
So both orientations give degenerate solutions unless e=1
But this contradicts the problem, which implies a **non−degenerate** solution.
So there's a **misunderstanding** of the configuration.
−−−
Wait: is the square CEFG drawn outward? Or can it be on the other side?
Wait: in both cases, C lies on segment DG, but maybe DG is a different segment?
Wait: point D is at (0,1), so DG goes from D through G, so if G is at (e,1), then DG is
from (0,1) to (e,1), which only reaches x=e. For C=(1,1) to lie on this segment, we need e
 \ge 1, which is not allowed.
But in **counterclockwise** case, G = (2 − e, 2e − 1)
Is it possible that in that case, DG contains C=(1,1)?
We only computed when C is exactly on segment: we found only e=1
But maybe **C lies on segment DG** does **not** require e=1?
Wait − we made an algebra mistake.
We had:
Parametrize segment from D=(0,1) to G=(2 − e, 2e −1)
We want point C=(1,1) to lie **on the segment** − not just on the ray − within the segment.

So solve:
x(t) = (1 − t)(0) + t(2 − e) = t(2 − e) = 1
So t = \frac{1}{2 − e}
Then y(t) = (1 − t)(1) + t(2e − 1)
Set y(t) = 1
So:
(1 − t)(1) + t(2e − 1) = 1
1 − t + t(2e − 1) = 1
```

```
- t + t(2e - 1) = 0
t( -1 + 2e - 1 ) = t(2e - 2) = 0

So again, either t=0 or 2e - 2 = 0 \Rightarrow e=1
Thus, only solution: e=1
So again - degenerated.
Thus, in both configurations, C lies on DG **only when e=1**.
So, **unless our assumption about the direction of square CEFG is wrong**, we have no
valid solution.
But the problem says "**C lies on segment DG**", and we are to assume e < 1
So what's wrong?
Ah - perhaps the square CEFG is not built externally from CE, but in another way?
Wait - maybe square CEFG is built **on side CE**, but using point C and E, and then
extending perpendicularly - but maybe the direction from C to G?
Wait - let's try a different labeling. Maybe CEFG means that from C to E, then E to F, F
to G, G to C. So the vertex G is such that \vec{CG} is obtained by rotating \vec{CE} by
90^\circ, or -90^\circ, depending on direction.
Wait - in a square CEFG, after C \to E, the turn from E to F is 90^\circ, so the direction
 CF?
No - we can get G as C + \text{rotation}(\vec{CE}, 90^\circ)?
Wait: in vector geometry, for a square CEFG, the point G such that C \to E \to F \to G \to
 C, then vector \vec{EG} is a 90^\circ rotation of \vec{EC}? Let's derive a formula.
Note: In square CEFG, vector \vec{EF} = R_{90^\circ}(\vec{CE}) and \vec{FG} = R_{90^\circ
}(\vec{EF}) = R_{180^\circ}(\vec{CE}) = -\vec{CE}
So:
- From C to E: \vec{v} = \vec{CE}
- E to F: R( \vec{v} )
- F to G: - \vec{v}
- G to C: -R(\vec{v})
So G = F + (-\vec{v}) = (E + R(\vec{v})) - \vec{v}
So:
G = E + R(\vec{v}) - \vec{v}

Let me test this.
We have \vec{v} = \vec{CE} = (0, e - 1)
Let R_{90^\circ} be **counterclockwise**: (x,y) \mapsto (-y, x)
Then R(\vec{v}) = R(0, e-1) = (-(e-1), 0) = (1 - e, 0)
Then G = E + (1 - e, 0) - (0, e - 1) = (1, e) + (1 - e, 0) - (0, e - 1)
= (1 + 1 - e, e) - (0, e - 1) = (2 - e, e) - (0, e - 1) = (2 - e, e - (e - 1)) = (2 - e,
1)
```
Wait - this is different from earlier!
```
So in this direction, G = (2 - e, 1)
Similarly, for clockwise rotation:
R_90^circ(\vec{v}) = (e - 1, 0) - because clockwise of (0, e-1) is (e-1, 0)
Then G = E + (e - 1, 0) - (0, e - 1) = (1, e) + (e - 1, 0) - (0, e - 1) = (1 + e - 1, e) -
 (0, e - 1) = (e, e) - (0, e - 1) = (e, e - (e - 1)) = (e, 1)
Wait - this is the same as before!
So in counterclockwise case, G = (2 - e, 1), not (2 - e, 2e - 1)
```
That was a mistake in previous work.So let's correct that.
```
---
### Revised: Correct rotation to get point G
Let's use:
G = E + R(\vec{CE}) - \vec{CE}, where R(\vec{v}) is a 90^\circ rotation.
- Case 1: Counterclockwise rotation
\vec{CE} = (0, e - 1)
R_{90^\circ}(\vec{CE}) = (-(e - 1), 0) = (1 - e, 0)
Then:
G = E + (1 - e, 0) - (0, e - 1) = (1, e) + (1 - e, 0) - (0, e - 1)
= (1 + 1 - e, e) + (0, -e + 1)
= (2 - e, e - e + 1) = (2 - e, 1)

So, G = (2 - e, 1)
- Case 2: Clockwise rotation
\vec{CE} = (0, e - 1)
R_{90^\circ}(\text{clockwise}) (0, e - 1) = (e - 1, 0)
```

```
Then:
G = E + (e − 1, 0) − (0, e − 1) = (1, e) + (e − 1, 0) − (0, e − 1) = (1 + e − 1, e) + (0,
−e + 1) = (e, e − e + 1) = (e, 1)
So again, same as before.
Thus corrected:
| Orientation | G |
|------------|-----|
| Counterclockwise | (2 − e, 1) |
| Clockwise | (e, 1) |
Now check the condition that **C = (1,1) lies on segment DG**.
Recall D = (0,1), C = (1,1), both on line y=1
---
#### Case 1: Counterclockwise square − G = (2 − e, 1)
Now D = (0,1), G = (2 − e, 1), both on horizontal line y=1
Segment DG is from x=0 to x=2 − e
We need (1,1) to lie on segment DG, i.e., 0 \le 1 \le 2 − e
So 1 \le 2 − e \Rightarrow e \le 1
Which is always true since e \in [0,1]
And 1 \ge 0 − always true.
So as long as 2 − e \ge 1, i.e., e \le 1, C lies **between** D and G?
Wait: (1,1) lies between D=(0,1) and G=(2 − e, 1) if 1 lies between 0 and 2 − e
So yes, when 2 − e \ge 1 -> e \le 1, which is always true.
But for the point to be **on segment**, we need 0 \le 1 \le 2 − e
So 1 \le 2 − e \Rightarrow e \le 1 − always true for allowed e
And 1 \ge 0 − trivial
So for **all** e \in [0,1], C = (1,1) lies on **segment DG**?
Wait − only if 2 − e \ge 1, which is e \le 1, true, and 1 is in [0, 2−e]
Yes. So every e \in [0,1] satisfies C \in DG
But is this actual?
For example: e = 0, E = (1,0) = B, G = (2 − 0, 1) = (2,1)
Then DG: D=(0,1) to G=(2,1) − so yes, C=(1,1) is midpoint.
For e = 1, E = C = (1,1), G = (1,1) − so G = C, and DG from (0,1) to (1,1) − so C is
endpoint − still on segment.
So **in counterclockwise case**, for **all** e \in [0,1], point C lies on segment DG
That **solves** the earlier issue − the constraint is **satisfied for all e** in this
orientation.
Now, in clockwise case: G = (e, 1)
So D = (0,1), G = (e,1), so segment from x=0 to x=e
We want C = (1,1) to lie on this − requires e \ge 1, again only when e=1
So only degenerate.
Thus, **only valid configuration is counterclockwise**, with G = (2 − e, 1), e \in (0,1)
---
### Step 4: Apply the collinearity: A, D, and O are collinear, where O is circumcenter of
triangle AEG
We have:
− A = (0,0)
− E = (1,e)
− G = (2 − e, 1)
We want circumcenter O of triangle AEG to lie on line AD
Now, AD is from A=(0,0) to D=(0,1), so it's the **vertical line x = 0**
So, the circumcenter O of triangle A,E,G lies on x=0
We now compute the circumcenter of triangle with vertices:
− A(0,0)
− E(1,e)
− G(2 − e, 1)
The circumcenter is the intersection of **perpendicular bisectors**
---
#### Step 4.1: Find perpendicular bisector of AE
− Midpoint of AE:
M_{AE} = \left( \frac{0+1}{2}, \frac{0+e}{2} \right) = \left( \frac{1}{2}, \frac{e}{2} \
right)

− Direction vector of AE: (1,e)
− So perpendicular direction: (−e, 1) or (e, −1) − dot product 1(−e) + e(1) = −e + e = 0
So perpendicular bisector has slope −\frac{1}{\text{slope of } AE} = −\frac{1}{e} (if e \
```

```
ne 0)
Slope of AE:  \frac{e - 0}{1 - 0} = e , so perp slope is  -\frac{1}{e}
So perp bisector: passes through (\frac{1}{2}, \frac{e}{2}), slope -\frac{1}{e}
Equation:
y - \frac{e}{2} = -\frac{1}{e} \left(x - \frac{1}{2} \right)

---
#### Step 4.2: Perpendicular bisector of AG
- A = (0,0), G = (2 - e, 1)
- Midpoint M_{AG} \left( \frac{0 + 2 - e}{2}, \frac{0 + 1}{2} \right) = \left( \frac{2 -
 e}{2}, \frac{1}{2} \right)
- Direction vector: (2 - e, 1)
- Perpendicular direction: (-1, 2 - e) or (1, -(2 - e)), since dot product: (2 - e)(-1) +
(1)(2 - e) = - (2 - e) + (2 - e) = 0
Take direction ( -1, 2 - e )
Slope of AG:  \frac{1 - 0}{2 - e - 0} = \frac{1}{2 - e}
So perp slope:  - (2 - e)
Thus, equation:
y - \frac{1}{2} = - (2 - e) \left( x - \frac{2 - e}{2} \right)

---
Now, the circumcenter O lies at intersection of these two perpendicular bisectors, and we
require x = 0
So plug x = 0 into first equation and solve for e, using the two equations.
---
**Step 4.3: Plug x=0 into equation of perp bisector of AE**
Equation:
y - \frac{e}{2} = -\frac{1}{e} \left( 0 - \frac{1}{2} \right) = -\frac{1}{e} \left( -\frac
{1}{2} \right) = \frac{1}{2e}

So:
y = \frac{e}{2} + \frac{1}{2e}

So the circumcenter has coordinates O = (0, \frac{e}{2} + \frac{1}{2e})
Now, we must also have this point satisfy the perp bisector of AG
So plug x=0, y = \frac{e}{2} + \frac{1}{2e} into the equation of the perp bisector of AG:
Equation:
y - \frac{1}{2} = - (2 - e) \left( x - \frac{2 - e}{2} \right)

Set x = 0:
Left side:  y - 1/2 = \left( \frac{e}{2} + \frac{1}{2e} \right) - \frac{1}{2} = \frac{e
}{2} - \frac{1}{2} + \frac{1}{2e}
Right side:  - (2 - e) \left( 0 - \frac{2 - e}{2} \right) = - (2 - e) \left( -\frac{2 - e
}{2} \right) = (2 - e) \cdot \frac{2 - e}{2} = \frac{(2 - e)^2}{2}
Set both equal:

\frac{e}{2} - \frac{1}{2} + \frac{1}{2e} = \frac{(2 - e)^2}{2}

Multiply both sides by 2:

e - 1 + \frac{1}{e} = (2 - e)^2

Compute right-hand side:
(2 - e)^2 = 4 - 4e + e^2
So:
e - 1 + \frac{1}{e} = 4 - 4e + e^2

Bring all terms to one side:
e - 1 + \frac{1}{e} - 4 + 4e - e^2 = 0

Simplify:
(5e - 5) - e^2 + \frac{1}{e} = 0

Multiply both sides by e to eliminate denominator:
e(5e - 5) - e^3 + 1 = 0
```

```
5e^2 - 5e - e^3 + 1 = 0
- e^3 + 5e^2 - 5e + 1 = 0

Multiply by -1:
e^3 - 5e^2 + 5e - 1 = 0

We now solve:
e^3 - 5e^2 + 5e - 1 = 0

Try rational roots: candidates +/-1
- e=1: 1 - 5 + 5 - 1 = 0 -> **yes!**
Factor out (e - 1)
Use polynomial division or synthetic division:
Divide e^3 - 5e^2 + 5e - 1 by e - 1
- Coefficients: 1 | -5 | 5 | -1
- Bring down: 1
- Multiply: 1 \cdot 1 = 1, add to -5 -> -4
- -4 \cdot 1 = -4, add to 5 -> 1
- 1 \cdot 1 = 1, add to -1 -> 0
So quotient: e^2 - 4e + 1
Thus:
(e - 1)(e^2 - 4e + 1) = 0
Solutions:
- e = 1
- e = \frac{4 \pm \sqrt{16 - 4}}{2} = \frac{4 \pm \sqrt{12}}{2} = \frac{4 \pm 2\sqrt
{3}}{2} = 2 \pm \sqrt{3}
Now 2 + \sqrt{3} \approx 2 + 1.732 = 3.732 > 1
2 - \sqrt{3} \approx 2 - 1.732 = 0.268 \in (0,1)
So only two valid solutions in [0,1]: e = 1, e = 2 - \sqrt{3}
Now e = 1 gives degenerate square: E = C, and G = (2 - 1,1) = (1,1), so square collapses.
We discard it.
So valid solution: e = 2 - \sqrt{3}
We now compute FG, which is the side length of square CEFG
Now, CEFG has side CE
C = (1,1), E = (1,e)
So distance:

CE = |1 - e| = 1 - e \quad (\text{since } e < 1)

We have e = 2 - \sqrt{3}
So:
1 - e = 1 - (2 - \sqrt{3}) = -1 + \sqrt{3} = \sqrt{3} - 1

Thus:
FG = CE = \sqrt{3} - 1
Wait - but earlier guess was sqrt2 - this is different.  Is that correct?
Wait: confirm the side length: yes, FG = CE, since CEFG is a square.
We have e = 2 - \sqrt{3} \approx 0.2679, 1 - e \approx 0.7321, and \sqrt{3} - 1 \approx
1.732 - 1 = 0.732 - yes.
So FG = \sqrt{3} - 1
But wait, is this the required answer?
Let me double-check all steps.
### Verification:
We assumed counterclockwise square with G = (2 - e, 1)
Validated that C = (1,1) on segment DG (yes)
Computed circumcenter: from perp bisector of AE, got x=0 at y = \frac{e}{2} + \frac{1}{2e}
Then required it to satisfy equation from perp bisector of AG, derived cubic: e^3 - 5e^2 +
 5e - 1 = 0
Roots: e=1, e = 2 \pm \sqrt{3}
Only 2 - \sqrt{3} \in (0,1)
Then CE = 1 - e = 1 - (2 - \sqrt{3}) = \sqrt{3} - 1
So FG = \sqrt{3} - 1
But wait - is this the simplest form?
Yes.
Now, is this value consistent?
```

```
Check if the circumcenter is well-defined.
For e = 2 - \sqrt{3}, all points are distinct: A, E, G not collinear?
Check: three points A(0,0), E(1,e), G(2-e,1)
They form a triangle? Yes, since not collinear.
And O is at x=0, meaning on line AD - as required.
So all conditions are satisfied.
---
Earlier summary said the only plausible symmetric value is sqrt2, but clearly that's not
the case - our derivation gives a clean algebraic solution.So we reject that guess.
Thus, correct answer is  \boxed{\sqrt{3} - 1}
Wait - is that possible? Let me compute approximate values:
- AB = 1
- CE = \sqrt{3} - 1 \approx 0.732
- So FG = \sqrt{3} - 1
Yes.
But the problem says "compute FG", and we have a clean expression.
Alternatively, could we have expressed in another form?
Alternatively, is there symmetry?
Note that in the expression:
e = 2 - \sqrt{3}
\Rightarrow 1 - e = \sqrt{3} - 1

Yes.
So final answer:
\boxed{\sqrt{3} - 1}
But let me just check - perhaps the square is measured differently?
Wait - the side of square CEFG is CE, which is distance between C and E.
C = (1,1), E = (1,e), so vertical segment of length 1 - e
Yes.
So side length FG = CE = 1 - e = \sqrt{3} - 1
All conditions are satisfied.
We tested the configuration: counterclockwise square, C on DG (since DG goes from D=(0,1)
to G=(2-e,1), with 2-e = 2 - (2 - \sqrt{3}) = \sqrt{3} \approx 1.732 > 1, so C=(1,1) is
strictly between D and G - yes.
E=(1,e) with e=2-\sqrt{3} \approx 0.267, so E lies on BC, from B=(1,0) to C=(1,1) - yes.
Circumcenter lies on AD - we derived it.
So all constraints satisfied.
---
### Conclusion
The previous attempt failed because:
- It incorrectly assumed the square orientation and coordinates.
- It used the wrong location for point G.
- It minimized over degenerate cases.
We corrected the configuration by properly applying rotation to get point G, and verified
that only the counterclockwise orientation satisfies all constraints.
We solved the collinearity condition and obtained a valid non-degenerate solution.}*)

Final answer:
\boxed{\sqrt{3} - 1}
```

# Q. RC Prompts

```
You are given a maths problem. You may also be given a summary of a previous attempt to solve it. This previous attempt may or may not be
correct.

### PROBLEM
{problem}

### SUMMARY OF PREVIOUS ATTEMPT
{curr_summary}

### INSTRUCTIONS
If no summary of a previous attempt is provided, solve the problem from scratch.

If a summary of a previous attempt is provided, your task is to improve upon this attempt. You should rely on this summary to guide your
thinking. Some examples of strategies you could use include:
- Verifying the previous solution.
- Proving the result in a different way.
- Finding alternative problem-solving strategies.
- Continuing from where the previous solution left off, assuming that the previous solution is incomplete.

Reason step-by-step and return your final answer in \\boxed{{}}.
```

*Figure 16.* **Summary-conditioned reasoning prompt** $\mathcal{I}_R$.

```
You are given a maths problem and a candidate solution to it. You may also be given a summary of a previous candidate solution to the problem.
If this is provided, you may assume that the current candidate solution was generated conditioned on the summary of the previous candidate
solution.
Your task is to write a summary of the current candidate solution.

The new summary you generate should possess the following characteristics:
- It should provide a detailed overview of what occurred in the current candidate solution. This may include a summary of the high-level
problem-solving strategy, a description of theorems used, verification attempts, calculations and logical deductions etc.
- It should summarize the current candidate solution in light of any previous summaries, if provided. We should be able to understand the
relationship between the previous solution and the current solution by reading the summary. Make sure any important information contained in the
existing summary is retained in the new one.
- It should be no more than two paragraph long and written in paragraph form, without headers or subheaders.
- It should be written in the first person, as if though it is being written by the person solving the problem.
- The candidate solution may not be complete. In this case, the summary should still attempt to summarize the partial solution.

IMPORTANT: Do not under any circumstances add any additional reasoning not contained in the latest reasoning step. Your task is only to
summarize what is given to you.

### PROBLEM
{problem}

### EXISTING SUMMARY
{existing_summary}

### LATEST CANDIDATE SOLUTION
{reasoning}
```

*Figure 17.* **Summarization prompt** $\mathcal{I}_S$.

```
You are given a maths problem. You may also be given a previous attempt to solve it. This previous attempt may or may not be correct.

### PROBLEM
{problem}

### PREVIOUS ATTEMPT
{prev}

### INSTRUCTIONS
If no previous attempt is provided, solve the problem from scratch.

If a previous attempt is provided, start by providing feedback on this solution. Then refine the solution based on the feedback.

Reason step-by-step and return your final answer in \\boxed{{}}.
```

*Figure 18.* **Self-refinement** $\mathcal{I}_{\text{refine}}$ **prompts.**

```
You are given a maths problem. You may also be given a previous attempt to solve it. This previous attempt may or may not be correct.

### PROBLEM
{problem}

### PREVIOUS ATTEMPT
{prev}

### INSTRUCTIONS
If no previous attempt is provided, solve the problem from scratch.

Review the previous attempt and identify possible mistakes. Afterwards, correct those mistakes and return the corrected answer.

Reason step-by-step and return your final answer in \\boxed{{}}.
```

*Figure 19.* **Self-verification** $\mathcal{I}_{\text{verify}}$ **prompt.**

```
Answer Only

- You should extract only the final, closed-form answer provided in the current solution. This is typically contained inside \\boxed{{}}. Do
not include any working or other logic.
- If no final answer is provided, just say "No answer available".

1 - 2 Sentences

- It should provide an overview of the approach taken in the candidate solution and in the previous summary.
- It should be brief, no more than 1 to 2 sentences long.

1 Paragraph

- It should provide a succinct overview of what occurred in the current candidate solution. This may include a summary of the high-level
problem-solving strategy, a description of theorems used, verification attempts, calculations and logical deductions etc.
- It should be no more than one paragraph long and written in paragraph form, without headers or subheaders.

2 Paragraphs

- It should provide a detailed overview of what occurred in the current candidate solution. This may include a summary of the high-level
problem-solving strategy, a description of theorems used, verification attempts, calculations and logical deductions etc.
- It should be no more than two paragraphs long and written in paragraph form, without headers or subheaders.

Multiple Paragraphs

- It should provide a highly detailed overview of what occurred in the current candidate solution. The summary should omit any intermediate
calculations (arithmetic, algebra etc.) but should describe all other steps, including failed or incorrect steps.
- It should be written in paragraph form, without headers or subheaders. It is expected to be multiple paragraphs long and highly detailed.
```

*Figure 20.* **Prompts from the summary length experiments in Section 5.2.** The instructions are inserted into the summarization prompt $\mathcal{I}_S$ in order to control the level of detail in the resulting summaries. The default level of detail is "2 paragraphs".

```
You are given a maths problem, a summary of a previous attempt to solve the problem, and the reasoning employed in a new attempt to solve it.
You are tasked with assessing how the problem-solving strategy of the new attempt is influenced by the previous attempt.

We identify these possible influences:
A. Verification: The new solution attempts to directly verify the previous solution and its logic.
B. Exploration: The new solution, informed by the previous solution, deliberately explores a different strategy to solve the problem.
C. Refinement: The new solution explicitly acknowledges the previous solution and walks through the same problem-solving steps in order to refine
it and build further confidence in the answer.
D. None: The new solution does not directly depend on or reference the previous solution, and appears to be an independent attempt at solving the
problem.

More than one influence may be present in the new attempt - select all the apply (A - D). D should only be selected on its own.
Provide an explanation before returning your final assessment (A-D) in \\boxed{{}} (e.g. \\boxed{{B}}). If multiple influences are present,
separate them with commas (e.g. \\boxed{{A, C}}).

### PROBLEM
{problem}

### SUMMARY OF PREVIOUS ATTEMPT
{summary}

### CURRENT ATTEMPT
{reasoning}
```

*Figure 21.* **Annotation prompt used by Qwen3-80B-Next-Instruct in Section 5.2.**

