# OpenReview forum: "Reasoning Cache: Continual Improvement Over Long Horizons via Short-Horizon RL"
_ICML.cc/2026/Conference — ICML 2026 regular_

### Official Review · Reviewer_Bi8D · 2026-03-05

**Soundness:** 3
**Presentation:** 2
**Significance:** 3
**Originality:** 3
**Overall Recommendation:** 5
**Confidence:** 3

**Summary:**

Increasing the test-time computation to improve LLMs’ performance can be achieved by a technique called reasoning, i.e., “first think and then answer.” This paper proposes an alternative called “reasoning cache” (RC). Specifically, it:

- proposes a decoding strategy that iteratively summarizes LLMs’ generations, and then let them generate based only on the summarization rather than the original long and wordy content;

- proposes a training pipeline to internalize this idea for instruction LLMs.

The logic to justify this idea is:

(1) Reasoning with the summarization/cache (RC) enables test-time computation: instruction models can increase their performance with more summarization rounds (Figure 2(a)).

(2) RC is better than simply extending the decoding token budget (Figure 4).

(3) RC is better than other test-time computation solutions such as “thinking” (RL with a thinking template), “self-refine” (another iterative decoding approach), and others.

**Compliance With Llm Reviewing Policy:**

Affirmed.

**Final Justification:**

The previous recommendation was "weak rejection", as it was mainly based on my concern about incorrect numbers: the sum of some percentages in a figure did not equal 100. In the rebuttal, the authors addressed this concern; therefore, I am willing to revise my recommendation to my original inclination: "accept".

**Key Questions For Authors:**

**Q1**
Page 4, Finding 4: How can you conclude that Qwen3-4B-Instruct-2507 is a good instruction follower, while the thinking mode is not? To be honest, I feel this claim is unscientific and unnecessary.

**Q2**
Page 5, Figure 3 (Middle): Please report the accuracy (not only the gain) in the appendix.

**Q3**
Page 5, Figure 3 (Right): Please check the sum of these bars. This is a very serious flaw.

I would like to give a commendation of **Weak Reject** to reflect my concerns. If the authors could address them, at least the incorrect numbers in Figure 3 (Right), I will consider increasing my score.

**Limitations:**

yes

**Strengths And Weaknesses:**

**Soundness, Good**

The logic of this paper is clear. It starts with the benefits of RC at decoding/testing time and then moves the comparison to trained models. The evaluation datasets are new and are after the cut-off time of both the model and the training set. The evidence supporting their claims comes not only from comparison results, but also from in-depth analyses such as the type of summarization (Figure 3, right) and the pass@k evaluation (Figure 5).


**Presentation, Fair to Good**

Some paragraphs are wordy. The most crucial feeling to me is that the authors want to tell everything in detail to the readers, and this makes the paper a bit heavy to me. But I acknowledge that the clarity of this work is good.

**Significance, Good**

This paper proposes an alternative way to enable test-time scaling (rather than “thinking”). It is a good attempt. The idea of reasoning cache is very promising, since the context length for training and testing can be very different. The proposed RC could, as evidenced by the paper’s results (especially the enlarged margins in Figure 4), better exploit the token budget.

**Originality, Good (low confidence)**

I am not in this area, but I am familiar with reasoning LLMs. To the best of my knowledge, this idea is novel.

---

> ### Author Rebuttal · Authors · 2026-03-31
>
> Thank you for your review of our paper! We are glad that you find the idea novel and believe that the paper is sound and significant, and the clarity is good. We address your questions below and would appreciate it if you could kindly check the responses to see if they address your questions
>
> > Page 4, Finding 4: How can you conclude that Qwen3-4B-Instruct-2507 is a good instruction follower, while the thinking mode is not? To be honest, I feel this claim is unscientific and unnecessary.
>
> We would like to clarify that by using the phrase “good instruction follower” we intended only to identify our empirical observation that the base Qwen3-4B-Instruct-2507 model appears to use the summarization outputs more actively for subsequent reasoning and generates more detailed summaries than the base Qwen3-4B-Thinking-2507 model. Our hypothesis was that the Instruct model is better at executing on its instructions (both the summarization and summary-conditioned generation instructions), as is the hallmark of good instruction following models: this seems like a natural explanation, as the instruct model is trained to excel at instruction-following tasks (as described on its Hugging Face page). This label is not meant to imply that the Instruct model will follow instructions of other types well in general.
>
> That said, we agree that this distinction may be unnecessary. While our experiments in Section 5 (Finding 4) show that RC decoding yields relatively larger gains with Qwen3-4B-Instruct-2507 than with Qwen3-4B-Thinking-2507, *RC still improves the thinking model's performance*. Moreover, after RC training, the thinking model's RC decoding performance improves significantly, indicating that it can also be trained to leverage an asymmetry between summarization and generation – see the first part of our response to Reviewer xYUz for details. We will make this much more explicit in our writing.
>
> > Page 5, Figure 3 (Middle): Please report the accuracy (not only the gain) in the appendix.
>
> We will include a version of this plot that reports the accuracy in the Appendix.
> An overview of the numbers: the “Inst/Inst” configuration improves from 39.8 to 55.8 at 384k reasoning tokens, “Think/Inst” improves from 56.9 to 65.8, while the “Think/Think” configuration from 55.4 to 58.3. The overall takeaway is that Inst/Inst yields the largest relative gains, although using the thinking mode for response generation + Instruct model for summarization (Think/Inst) yields better absolute performance.
>
> > Page 5, Figure 3 (Right): Please check the sum of these bars. This is a very serious flaw.
>
> We presume the concern is that the bars do not sum to 100. We actually do not expect the bars to sum to 100 because we allow the annotator model to select more than one option for every data point (see the prompt in Fig 21 in Appendix P). This is because summary-conditioned reasoning sometimes pursues more than one “strategy”: e.g. there are quite a few samples where the model both verifies past reasoning and also re-attempts problem solving with a new strategy.
>
>
> ### Updated IMO AnswerBench results
>
> Finally, we would like to take this opportunity to update our IMO-AnswerBench results (relevant to all reviewers). In our original submission, we reported IMO-AnswerBench results evaluated using math-verify [1]. We recently discovered that this approach underestimates accuracy, as math-verify penalizes certain valid answer formats, leading to correct responses with alternate answer formats being still marked as incorrect. We therefore report results using the official LLM judge-based evaluation protocol from [1] instead, replacing Gemini-2.5-Pro with Gemini-3-Flash. An overview of the updated main results is shown below, where we find the raw performance of all models to improve. Importantly, the new evaluation does not change any of our conclusions: **the relative ordering of all models is preserved, and all key findings continue to hold.**
>
> | Model | Avg. Score on IMO-AnswerBench |
> |-------|----------------:|
> | Qwen3-4B-Instruct-2507 [16k] | 40.9 |
> | Qwen3-4B-Instruct-2507 (RL, 32k) [32k] | 42.1 |
> | Polaris-4B [2] [90k] | 47.7 |
> | Qwen3-4B-Thinking-2507 [81k] | 53.8 |
> | | |
> | Self-Refine (base) | 45.8 |
> | Self-Verify (base) | 43.9 |
> | Self-Refine (trained) | 52.2 |
> | Self-Verify (trained) | 52.3 |
> | | |
> | Qwen3-4B-Instruct-2507 + RC | 46.3 |
> | Qwen3-4B-Instruct-2507 (RL, 32k) + RC | 53.4 |
> | **RCT-4B + RC** *(Ours)* | **58.0** |
>
>
> | Model | Avg. Score on IMO-AnswerBench |
> |-------|---------------------:|
> | Qwen3-4B-Instruct-2507 | 40.9 |
> | RCT-4B (16k) | 44.5 |
> | Polaris-4B | 47.7 |
> | Qwen3-32B | 49.2 |
> | Qwen3-80B-Next-Instruct | 51.4 |
> | Qwen3-30B-A3B-Instruct-2507 | 52.9 |
> | Qwen3-235B | 53.8 |
> | Qwen3-4B-Thinking-2507 | 53.8 |
> | Nemotron-3-Nano-30B-A3B | 57.9 |
> | **RCT-4B + RC (256k)** *(Ours)* | 58.0 |
> | gpt-oss-20B | 61.5 |
> | Qwen3-30B-A3B-Thinking-2507 | 65.9 |
>
> ---
>
> [1] github.com/huggingface/Math-Verify
>
> [2] arxiv.org/abs/2511.01846

---

> > ### Author Rebuttal · Reviewer_Bi8D · 2026-04-03
> >
> > Thank you for the clarification. I encourage you to revise the paper accordingly and am willing to raise my recommendation from a weak reject to accept.

---

> > > ### Author Response · Authors · 2026-04-05
> > >
> > > Thank you for your positive review, and for the helpful feedback! We will implement changes accordingly.

---

### Official Review · Reviewer_xUYz · 2026-03-13

**Soundness:** 3
**Presentation:** 3
**Significance:** 3
**Originality:** 3
**Overall Recommendation:** 4
**Confidence:** 2

**Summary:**

This paper focuses on a very important and interesting problem: it's difficult for models in the current RL training paradigm to reason and optimize under extrapolation with a limited training budget. By analyzing some of the potential problems with standard RL, the authors propose the Reasoning Cache (RC) approach, which summarizes the reasoning traces during long reasoning. Based on this, the authors also propose the RC training algorithm to train models to extrapolate with RC. The above algorithms effectively enhance the model reasoning performance under the limited token budget.

**Compliance With Llm Reviewing Policy:**

Affirmed.

**Final Justification:**

The author's response addresses my concerns, and I maintain my positive score.

**Key Questions For Authors:**

Please refer to the above weaknesses.

**Limitations:**

yes

**Strengths And Weaknesses:**

### **Strengths**
- This paper focuses on a very critical and meaningful limitation in reasoning tasks.
- Well-structured paper and detailed analyses. Results on a variety of experimental settings strengthen the conclusions.
- The Reasoning Cache algorithm is simple, well-motivated, and easy to follow. The authors conduct thorough and in-depth investigations based on this paradigm, developing it into an extrapolable and trainable reasoning mechanism.
### **Weaknesses**
- Only limited tasks are investigated in this paper. The experiments still focus on long reasoning tasks with clear answers and do not give an analysis of possible negative examples on other tasks. Thus, the current analyses still fail to prove that RC is a long reasoning extrapolation framework with sufficient generalizability.
- The results show that RC is more applicable to instruction-following models as opposed to highly specialized reasoning models. This reflects the limitations of the current methodology, and a more systematic analysis would be helpful.
- An LLM-based annotator is used in Section 5.3 to classify reasoning traces. While this may be valid, the lack of manual validation and example analysis significantly reduces its credibility and reliability.

---

> ### Author Rebuttal · Authors · 2026-03-31
>
> Thank you for your review of our paper!
>
> To address the first two weaknesses, we would like to share some **new results**. We trained Qwen3-4B-Thinking-2507 on olympiad proof problems from the Fineproofs [1] dataset, producing a model we call RCT-Proofs-4B. Notably, this task differs from our existing setup in two ways: the answers are open-ended proofs rather than short verifiable answers, and rewards are assigned by an LLM judge (gpt-oss-20b) using a rubric rather than by string matching. Due to time constraints, we performed Stage I training only, with 3 training turns and a reasoning length (H_R) of 50k tokens.
>
> We evaluate RCT-Proofs-4B on IMO-ProofBench [2] and report results after 8 RC turns. For comparison, we also report the performance of RCT-4B (16 RC turns, H_R = 16k) and other baseline models (e.g. the standard RL-trained model). Grading is done using Gemini-3.1-Pro (medium) with the official grading prompt, which scores proofs on a 7-point scale according to the judge rubrics, although we normalize and report scores on a 0-100 point scale. We report average scores over three repetitions.
>
> | Model | Avg. Score on IMO-ProofBench |
> |-|-:|
> | Qwen3-4B-Thinking-2507 | 14.3 |
> | Qwen3-4B-Thinking-2507 + RC | 15.9 |
> | Qwen3-4B-Thinking-2507 (RL, 50k) + RC | 24.0 |
> | **RCT-Proof-4B + RC** | **31.1** |
> | | |
> | Qwen3-4B-Instruct-2507 | 9.9 |
> | Qwen3-4B-Instruct-2507 + RC | 14.3 |
> | Qwen3-4B-Instruct-2507 (RL, 32k) + RC | 17.7 |
> | **RCT-4B + RC** | 22.4 |
>
> We also report the performance of RCT-Proofs-4B on IMO-AnswerBench.
> | Model | Avg. Score on IMO-AnswerBench |
> |-|-:|
> | Qwen3-4B-Thinking-2507 | 53.8 |
> | | |
> | Qwen3-4B-Thinking-2507 + RC | 57.6 |
> | Qwen3-4B-Thinking-2507 (RL, 50k) + RC | 60.8 |
> | | |
> | **RCT-Proof-4B + RC** | **65.3** |
>
> As can be seen in the tables above, RC training significantly improves model performance on IMO-ProofBench, both when we directly train the thinking model on proof problems (RCT-Proofs-4B) and when we train the instruct model on problems with short answers (RCT-4B). This generalization property holds the other way as well, with RCT-Proofs-4B demonstrating significant improvement on IMO-AnswerBench despite being RC-trained only on proof problems. Please see our reply to Reviewer Bi8D for a discussion of our updated IMO-AnswerBench results more generally.
>
> To address the listed weaknesses directly:
>
> > Only limited tasks are investigated in this paper. The experiments still focus on long reasoning tasks with clear answers…
>
> Our new experiments on olympiad-level proof problems directly address this. Unlike the tasks in our original submission, proof problems require long, open-ended answers without a clear closed-form solution, and are graded by an LLM judge using a rubric rather than by string matching. As shown in the tables above, RC training yields significant improvements on this task, with RCT-Proofs-4B achieving an average score of 31.1 on IMO-ProofBench compared to 14.3 for the base model. Moreover, this improvement transfers to IMO-AnswerBench (53.8 → 65.3), demonstrating that RC generalizes across task formats.
>
> > The results show that RC is more applicable to instruction-following models as opposed to highly specialized reasoning models…
>
> Our new results show that RC training is also effective for specialized reasoning models. Notice that RC decoding without training yields relatively modest improvements on the thinking model (14.3 → 15.9, +11%) compared to the instruct model (9.9 → 14.3, +44%), which mirrors our findings from Section 5 (Finding 4). However, training the thinking model to use RC leads to substantial gains (14.3 → 31.1, +118% on IMO-ProofBench, 53.8 → 65.3, +21% on IMO-AnswerBench). This suggests that while the thinking model is less able to exploit RC decoding out of the box, the asymmetry is large enough that targeted training on the thinking model can enable significant improvements. Note that our experiments in Section 5 were based largely on base models where we did find similar conclusions. That said, we will add these new results to the paper and make this argument clearer in both Sec. 5 and 7.
>
> > An LLM-based annotator is used in Section 5.3 to classify reasoning traces. While this may be valid, the lack of manual validation and example analysis significantly reduces its credibility and reliability.
>
> We manually validated a sample of the LLM annotations and generally found them to be accurate. We agree that manual analysis could strengthen this section: however, we note that the LLM-based annotation is used primarily to identify broad trends across a large number of reasoning traces, where manual annotation of the full dataset would be prohibitively expensive. The qualitative findings from Sec. 5.3 are also consistent with the quantitative results we observe throughout the paper, which provides additional support for their validity.
>
> ---
>
> [1] huggingface.co/datasets/lm-provers/FineProofs-RL
>
> [2] arxiv.org/abs/2511.01846

---

> > ### Author Rebuttal · Reviewer_xUYz · 2026-04-03
> >
> > Thank you for your reply. The author's response addresses my concerns, and I maintain my positive score.

---

> > > ### Author Response · Authors · 2026-04-05
> > >
> > > Thank you for your positive review, and we are glad that your concerns have been addressed.
> > >
> > > Please let us know if you have any more concerns about our work, or if there are any specific weaknesses you would like us to address.

---

### Official Review · Reviewer_hxpS · 2026-03-17

**Soundness:** 4
**Presentation:** 3
**Significance:** 4
**Originality:** 3
**Overall Recommendation:** 6
**Confidence:** 4

**Summary:**

RC is a simple test time scaling approach: replace a model's context with a summary of that context, which the model can then leverage to continue/refine/verify past reasoning while staying within a constant context window size. This approach outperforms competing approaches to scaling inference compute sequentially, and it is compatible with parallel scaling approaches, unlocking very strong performance. Moreover, performance improves when training models to reason based on summaries during RL.

**Compliance With Llm Reviewing Policy:**

Affirmed.

**Final Justification:**

The rebuttal reinforced my prior assessment and addressed my main concerns.

**Key Questions For Authors:**

My concerns mainly relate to secondary claims of the submission (e.g. the FrontierScience claim mentioned above) that might need to be rephrased or better supported. The FrontierScience point above is the most critical to address, followed by the first 3 points below. I would be happy to raise my score given reasonable responses to these 4 points. Addressing the remaining points would possibly help the paper but is not critical.

1. Line 431: is this true? It seems like the gap between regular RL and RC RL is constant across test compute values in Figure 6.

2. Figure 10 suggests replay buffer summaries from earlier training steps are removed as summaries from newer training steps are added. This seems like it makes the replay buffer only one step off-policy, but it seems to nonetheless have a big effect in Figure 6. Please let me know if I'm misunderstanding something here. If the above is correct, would increasing the off-policyness more help? E.g., by sampling many more summaries from the untrained model on the first training step.

3. What would Figure 3 middle look like with absolute accuracies instead of gains? Combining this with Table 1, it seems like the best approach is actually RC with the Thinking model, but this is not made clear in the discussion.

4. Line 266: it's unclear what is meant by “redundancy” in this context.

5. Is it worth discussing the connection of RC to context compaction? https://github.com/openai/codex/blob/95bdea93d2600aabef1b87ee5fab05a6022a7d45/codex-rs/core/templates/compact/prompt.md

6. Line 324: should this be T_train*H_R?

7. Are the memory states in Suzgun et al. (2025) also created via an LLM summary of its own prior outputs? I'm not sure if their work’s setting only differs from the submission’s because it doesn't train models to better utilize memory, as the submission's related work section suggests in its final paragraph.

**Limitations:**

Yes.

**Strengths And Weaknesses:**

### Soundness:

- The broad claims about effectiveness are very well supported by experimental results, and limitations are clearly laid out.

- Ablation studies support design choices of the RC scaling approach and its associated training method.

- Lines 97/369/344-right/etc.: I don't think this FrontierScience claim is well supported. RL training vs. RC training only induces a 0.6% gap in FS performance, which is likely not close to being statistically significant.

### Presentation

- The figures and contextualization with respect to prior work are excellent, and the approach is very well motivated (e.g., by laying out the need for extrapolation). Some things should be communicated more clearly, though.

- The opening paragraph of Section 4 seems contradictory with the success of s1/budget-forcing.

- Line 107 says prior work enables extrapolation, which is apparently contradicted later in the same paragraph when it is said that “these approaches do not improve” when test time distributions are outside the training support.

- It's unclear what dataset was used for the experiments in Table 2.

- Figure 5 might be clearer with shared y axes.

### Significance

- This paper will have a very high impact on areas of ML that relate to inference compute scaling. Particularly, I expect the community to further study and leverage the submission’s demonstrated improvements from iterative decoding based on summary and reasoning, training models to hone summary-conditioned reasoning, and incorporating such honed models into other test time scaffolds like RSA/DSM.

### Originality

- Leveraging summarization to expand effective test time compute without altering the context size is not profoundly novel – as the submission's related work should perhaps better clarify, other iterative test time scaling methods like RSA can leverage summarized reasoning from a prior iteration to improve reasoning in the current iteration (though RSA only explicitly instructs the model to aggregate past reasoning summaries, leveraging them to facilitate extended reasoning is possible too).

- The submission’s method avoids confounding factors like the parallel scaling axis, providing a unique focus on iterative summary alone and correspondingly original insights. The novel insights here rely on careful evaluation of how the proposed approach scales, how this scaling behavior improves when models are trained to do summary conditioned reasoning, and how such trained models enhance other test time scaling approaches (even without RC).

---

> ### Author Rebuttal · Authors · 2026-03-31
>
> Thank you for your positive review! We would first like to point you to our responses to Revs. xUYz and Bi8d, where we detail new results:
> - [xUYz] We RC-train a thinking model on proof problems, and demonstrate large gains.
> - [Bi8d] We update our IMO-AnswerBench results after improving our evaluation methodology.
>
> ---
>
> > I don't think this FS claim is well supported.
>
> To address this, we present new results on FS that establish the efficacy of RC, and showcase that when training on more challenging proof problems, the gap between RL and RC trained models grows as more test-time compute is provided.
>
> - We extended RC decoding from 16 to 32 turns. The goal here is to determine how extrapolating length further affects the gap between RL & RC.
> - We evaluated our proofs-trained model RCT-Proofs-4B on FS.
>
> | Model | Avg. Score on FS|
> |-|-:|
> | Inst | 23.3 |
> | Inst + RC | 29.5 |
> | Inst (RL) + RC (T = 16) | 33.0 |
> | Inst (RL) + RC (T = 32) | 32.0 |
> | RCT-4B + RC (T = 16) | 34.1 |
> | RCT-4B + RC (T = 32) | 36.1 |
> | Think | 25.7 |
> | Think + RC | 33.0 |
> | Think (RL) + RC | 32.0 |
> | RCT-Proofs-4B + RC | 37.3 |
>
> Extending decoding improves RCT-4B (34.1 → 36.1) but not the RL baseline’s (33.0 → 32.0) performance. RC training on the proofs-trained model yields improvement over the base model with RC decoding (33.0 → 37.3), whereas std. RL training does not (33.0 → 32.0). These results suggest that **RC training does teach the model to make use of summaries on FS, and the gains compound over turns, even if gains are smaller than for math (which perhaps is expected since FS is out-of-domain). We will adjust the writing to reflect this.**
>
> > Fig 10 suggests replay buffer summaries from earlier training steps are removed as summaries from newer training steps are added…
>
> We clarify that every time we generate new summaries, we remove previous summaries *corresponding to the same problem* from the buffer. However, because a given problem may not be sampled again for many train steps, the summaries are very off-policy by the time it is sampled again.
>
> > What would Fig 3 middle look like with absolute accuracies instead of gains?
>
> Please see our response to Rev. Bi8d.
>
> > Line 431: It seems like the gap between regular RL and RC RL is constant across test compute values.
>
> The gap between regular RL and RC RL is indeed roughly constant in this experiment. We will revise the writing.
>
> That said, we dug deeper into this by repeating this study with RCT-Proofs-4B and the corresponding RL model on IMO-ProofBench, a more difficult benchmark that requires longer responses than answer-based questions. Note that the gap between the two **does increase** w/ turns (scores out of total 7). We believe this is because producing the correct proof over long lengths requires the model to faithfully follow + build on the summary, which only RC training trains it to do (and not RL).
>
> | Turn | RL-trained | RC-trained | Delta |
> |-:|-:|-:|-:|
> | 0 | 0.84 | 0.92 | +0.08 |
> | 3 | 1.16 | 1.33 | +0.17 |
> | 7 | 1.20 | 1.39 | +0.19 |
> | 11 | 1.20 | 1.47 | +0.27 |
> | 15 | 1.24 | 1.54 | +0.30 |
>
> > The opening paragraph of Section 4 seems contradictory with the success of s1.
>
> This paragraph is not contradictory to s1: budget forcing indeed produces gains (Tab. 1), but these are modest and saturate quickly (~2-4x the train budget). Sec. 4 provides a conceptual explanation: s1 extends reasoning by appending “Wait” tokens, which pushes generation into regions far from the train distribution. RC avoids this by keeping each step within the train regime, which enables extrapolation to longer horizons. We will revise Sec. 4 to make clear that our claim is about why s1’s gains plateau, not that such gains do not exist.
>
> > Line 107 says prior work enables extrapolation, which is apparently contradicted later…
>
> Our survey of previous extrapolation methods indicate that they enable extrapolation for a few times the train budget (line 107, first col.), but generally struggle to improve further (line 77, second col.). In contrast, RC can extrapolate to an order of magnitude more reasoning tokens. We will make this argument clearer.
>
> > It's unclear what dataset was used for the experiments in Table 2.
>
> HMMT 2025 (Nov). We will add this to the paper.
>
> > Line 266: it's unclear what is meant by “redundancy” in this context.
>
> Redundancy refers to the fact that many tokens are unnecessary for further reasoning as their contributions are captured by other tokens (line 169), and can thus be summarized without detrimental impact. The idea here specifically is that when H_R is too small, we risk many steps being terminated very early, before redundancy can arise (i.e. because key steps, calculations etc. are not yet complete).
>
> On **related work** (context compaction, Suzgun et al. 2025, comparison w. RSA): we will amend our related work discussion based on your feedback. Please let us know if you would like us to elaborate on any specific points in our follow-up discussion.

---

> > ### Author Rebuttal · Reviewer_hxpS · 2026-04-03
> >
> > All my points were well addressed. I will raise my score to 6.
> >
> > Please improve the discussion of Figure 3 middle (e.g. referencing the absolute performance plot added in the appendix) and of the meaning of redundancy.

---

> > > ### Author Response · Authors · 2026-04-05
> > >
> > > Thank you for your very thorough and thoughtful review, and for the helpful feedback! We will implement changes accordingly.

---

### Decision · Program_Chairs · 2026-04-30

**Decision:**

Accept (regular)

**Comment:**

This paper addresses the pivotal challenge of length extrapolation in reinforcement learning for long-horizon reasoning. The authors propose Reasoning Cache (RC), an iterative algorithm that periodically compresses reasoning traces into summaries, enabling models to extend their reasoning far beyond training budgets while maintaining alignment with the training distribution.
While initial reviews raised concerns regarding the diversity of evaluation datasets, specific numerical results, and presentation clarity, the authors successfully addressed these issues through comprehensive rebuttals and supplementary experiments. The reviewers acknowledged these clarifications and subsequently raised their scores. Given its solid contribution to the field of test-time compute scaling, I recommend the paper for acceptance.